# Three-degrees-of-freedom orientation manipulation of small untethered robots with a single anisotropic soft magnet

Heng Wang [1,2] ✉, Junhao Cui [1,2], Kuan Tian [1] & Yuxiang Han[1]

Magnetic actuation has been well exploited for untethered manipulation and locomotion of small-scale robots in complex environments such as intra-corporeal lumens. Most existing magnetic actuation systems employ a permanent magnet onboard the robot. However, only 2-DoF orientation of the permanent-magnet robot can be controlled since no torque can be generated about its axis of magnetic moment, which limits the dexterity of manipulation. Here, we propose a new magnetic actuation method using a single soft magnet with an anisotropic geometry (e.g., triaxial ellipsoids) for full 3-DoF orientation manipulation. The fundamental actuation principle of anisotropic magnetization and 3-DoF torque generation are analytically modeled and experimentally validated. The hierarchical orientation stability about three principal axes is investigated, based on which we propose and validate a multi-step open-loop control strategy to alternatingly manipulate the direction of the longest axis of the soft magnet and the rotation about it for dexterous 3-DoF orientation manipulation.

Small-scale untethered robots, ranging from pill-sized capsule robots to milli-scale and even micro-scale robots, have demonstrated great potential in various applications such as micro-manipulation[1–3], pipeline inspection[4,5], and medical operations[6–10]. Small untethered robots can navigate through tortuous pathways in complex environments and access hard-to-reach target places with high precision due to their tiny size and high maneuverability. Especially in minimally invasive medical procedures, untethered robots are expected to deliver drugs and implants[11–13], examine and diagnose potential diseases[14–16], and perform surgical tasks[17–19] in narrow lumens and cavities inside the human body that are impossible or difficult to access using existing medical devices. Employment of small untethered robots in medical operations can avoid large open incisions and thus bring benefits of shortened recovery time and reduced medical complications and risks. In the last two decades, a few untethered medical robots have been put into clinical use such as capsule endoscopes[20], which use an embedded camera to image the gastrointestinal tracts for diagnostic purposes. Compared with tethered devices such as flexible continuum robots and catheters[21–23], untethered robots have better maneuverability and dexterity since they are not constrained by the tether. Furthermore, tethered robots cannot serve as permanent or temporary implants[7].

To enable mobility and manipulation of untethered robots, an actuation system needs to be developed based on either self-propulsion methods or external actuation methods. Self-propulsion methods integrate power sources, motors, and micro-mechanisms (e.g., legs[24,25], fins[26], treads[27], balloons[28] etc.) onboard the untethered robot for actuation and locomotion. However, it is not feasible to integrate these self-propulsion components into robots in the sub-millimeter scale due to the limited internal space. Instead, external power from magnetic fields[15,16,29–33], light[34,35], acoustic waves[36], and chemicals[37] can be applied to drive small robots. Except for magnetic fields, other external actuation methods are normally used for actuation of micro- and nano-scale robots due to their limited driving force. Magnetic fields can permeate the human body safely without the occlusion problem and can exert strong forces and torques on robots with an onboard magnetic object over a long range. Therefore,

[1]Shien-Ming Wu School of Intelligent Engineering, South China University of Technology, Guangzhou, Guangdong, China. [2]These authors contributed equally: Heng Wang, Junhao Cui. ✉e-mail: wanghengscut@scut.edu.cn

magnetic actuation emerges as an ideal remote actuation method for locomotion and manipulation of small untethered robots across different scales.

Magnetic actuation systems can be categorized as direct actuation systems and indirect actuation systems assisted by secondary propulsion mechanisms[32]. In direct magnetic actuation systems, the robot embedded with a magnet is directly driven by the magnetic force and torque generated by the interaction between the onboard magnet and an externally applied magnetic field, with no need for additional onboard mechanisms. The external magnetic field can be generated by either an array of electromagnetic coils[38–41] or a permanent magnet[42–44]. The direction and strength of the applied field and field gradients can be controlled by tuning the electric currents and/or pose of the magnetic sources. In indirect magnetic actuation systems, propulsion mechanisms, e.g., helical propellers[45], screw threads[46], and soft legs[47], are designed and integrated into the magnetic robot to transmit the power of the externally applied rotating magnetic field to oscillations of the propellers for locomotion. In this work, we are focused on the direct magnetic actuation method.

Existing magnetic actuation systems usually employ a single permanent magnet rigidly fixed to the small robot[39,42,46], which can generate magnetic forces in three directions and magnetic torques in two directions in a non-uniform magnetic field. However, no magnetic torque can be generated about the axis of the magnetic moment of the permanent magnet. Therefore, the degrees of freedom (DoF) of orientational manipulation are limited to two although 3-DoF translational manipulation is feasible. The lack of the 3rd DoF of orientation actuation limits the dexterity of manipulation and locomotion of untethered robots and impairs their capability of performing sophisticated tasks. Researchers (Diller et al., Giltinan et al., and Thornley et al.) have proposed two innovative approaches to remedy this deficiency. The first approach is to add one or two auxiliary magnets to generate magnetic torque about the 3rd DoF of orientation via a force couple[48,49]. The problem with this approach is that the use of multiple magnets increases the weight of the robot and requires a complicated assembly technique. Another approach is to use a single permanent magnet with a complicated anisotropic shape to generate the magnetic torque in the 3rd DoF, which can be considered as a continuous variant of the first approach[50]. A common problem with these two approaches is that the added torque in the 3rd DoF attenuates magnetic torques in the other 2 DoFs[49].

Unlike permanent magnets with fixed axis of magnetization, the magnetization of soft magnets varies with the applied magnetic field and thus soft magnets exhibit different dynamic behaviors in magnetic actuation. It has been pointed out by earlier literature that soft magnets have the potential for 6-DoF magnetic manipulation[32]. Although Abbott et al. derived a model of magnetic torque and force on an axially symmetric soft magnet that is capable of 2-DoF orientation manipulation[51], neither dynamic modeling nor experimental demonstration of 3-DoF orientation manipulation using a soft magnet was conducted.

Here we propose a magnetic actuation method that uses a single anisotropic soft magnet instead of a permanent magnet to enable full 3-DoF orientation manipulation of small untethered robots. This paper starts with modeling of the geometry-induced tri-axial anisotropic magnetization of a single soft-magnetic object, which is the fundamental mechanism for 3-DoF torque generation. Then tri-axial ellipsoids and elliptical cylinders are proposed to demonstrate the capability of 3-DoF torque generation. The 3-DoF magnetic torques are modeled and experimentally validated, which shows that the magnetic torque about the second-longest axis is the maximum and is equal to the sum of the torques about the other two axes. The analysis of orientation stability shows that the tendency of the three principal axes of the ellipsoidal soft magnet to align with the applied magnetic field increases with the axial length, with the longest axis having the strongest tendency to align with the field. Based on this hierarchical orientation stability among three principal axes, an open-loop control strategy for 3-DoF orientation manipulation is proposed using a pair of orthogonal magnetic fields applied alternatingly, with one field to adjust the direction of the longest axis (2-DoF orientation control) and the other one to adjust the rotation about the longest axis (1-DoF orientation control). Experiments show that the proposed orientation control method can achieve dexterous manipulation over any arbitrarily defined 3-DoF orientation paths. The investigation on the soft-magnet-based 3-DoF orientation manipulation in this work lays the foundation for full 6-DoF magnetic manipulation of untethered robots using a single magnetic object with a simple geometry.

## Results
### Magnetic manipulation of small robots with an on-board permanent magnet
Magnetic actuation has been well exploited to manipulate and navigate small-scale untethered robots through lumens inside the human body for therapeutic and diagnostic procedures (Fig. 1a). Most conventional magnetic actuation methods employ an array of electromagnetic coils or a manipulator-controlled permanent magnet to generate the desired magnetic field to drive the robot equipped with an on-board permanent magnet. The magnet can experience force and torque in an applied magnetic field, which attempts to translate or rotate the magnet to minimize the magnetic potential energy $-\mathbf{B} \cdot \mathbf{m}$[32], where $\mathbf{B}$ is the local magnetic field and $\mathbf{m}$ is the magnetic moment of the magnet. The magnetic force on a magnet depends on the spatial gradient of the local magnetic field (Fig. 1c, d), which is given by

$$f = \nabla(\mathbf{B} \cdot \mathbf{m}) = \left[\frac{\partial \mathbf{B}}{\partial x} \frac{\partial \mathbf{B}}{\partial y} \frac{\partial \mathbf{B}}{\partial z}\right]^T \mathbf{m} \qquad (1)$$

where $\mathbf{B}_{\nabla} = \left[\frac{\partial \mathbf{B}}{\partial x} \quad \frac{\partial \mathbf{B}}{\partial y} \quad \frac{\partial \mathbf{B}}{\partial z}\right]^T$ is the matrix of magnetic field gradients. It is noted that magnetic force cannot be generated on the magnet in a uniform magnetic field with zero magnetic field gradient. The magnetic torque on a magnet depends on the local magnetic field itself, which is given by

$$\boldsymbol{\tau} = \mathbf{m} \times \mathbf{B} = \begin{bmatrix} 0 & -m_z & m_y \\ m_z & 0 & -m_x \\ -m_y & m_x & 0 \end{bmatrix} \begin{bmatrix} B_x \\ B_y \\ B_z \end{bmatrix} \qquad (2)$$

It is shown that the magnetic torque is the cross product of the magnetic moment and the local magnetic field, and it always attempts to align the magnet with the field. Equation (2) implies that the magnetic torque on a permanent magnet is always perpendicular to the magnetic moment vector (Fig. 1b). Therefore, magnetic actuation systems using a single permanent magnet on the robot can never generate a torque about the axis of the magnetic moment, limiting orientation manipulation to only two DoFs. Due to the loss of the 3rd magnetic torque, only 5-DoF manipulation of the magnet can be achieved and the dexterity of magnetic manipulation is impaired in the permanent-magnet-based actuation systems. To meet this challenge, this paper proposes a solution that replaces the permanent magnet onboard the robot with a non-spherical soft magnet, which can achieve full 3-DoF orientation manipulation under anisotropic magnetization of an actuating magnetic field.

### Geometry-dependent anisotropic magnetization of the soft magnet
Ferromagnetic materials exhibit strong magnetism in the presence of an applied magnetic field and they can be categorized as hard-

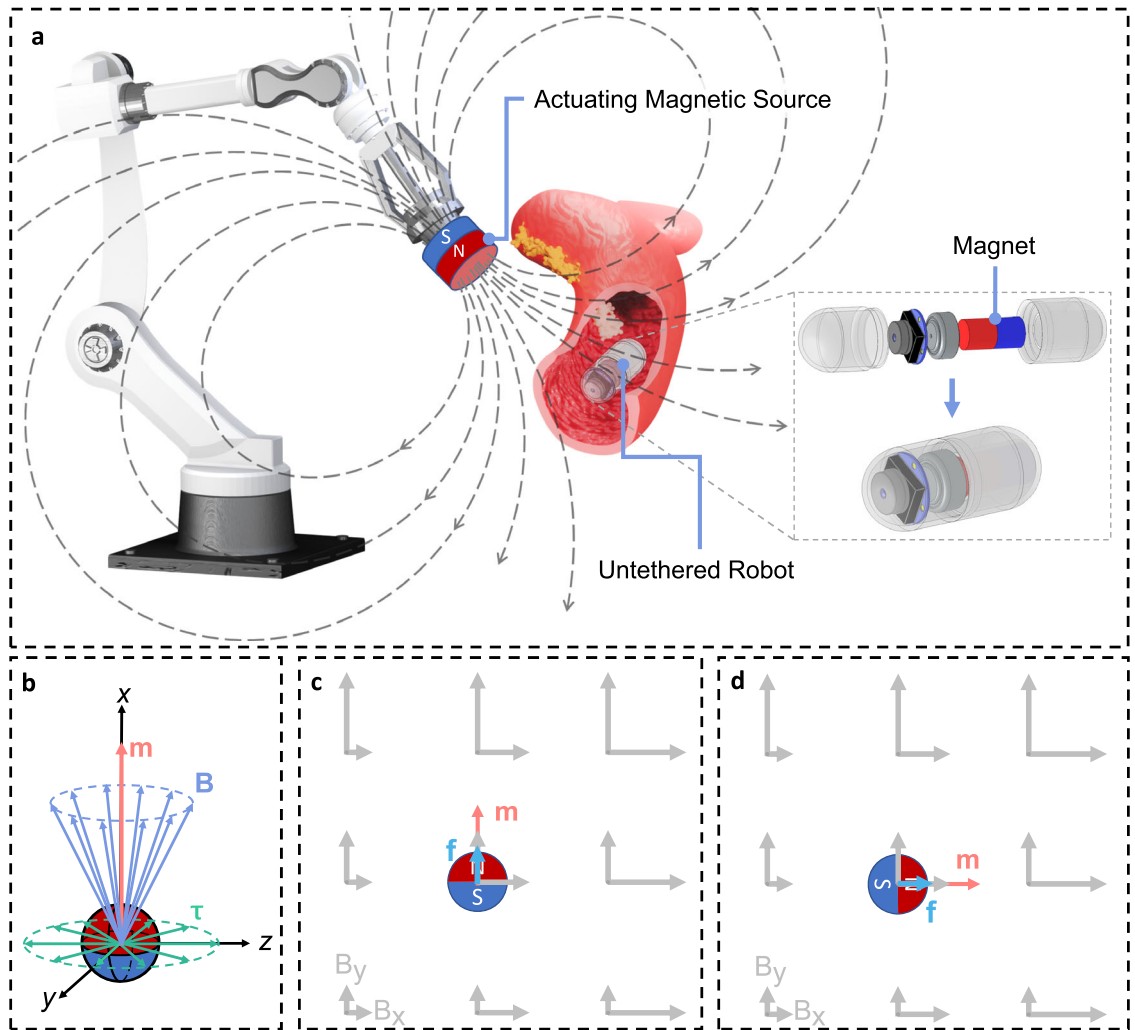

**Fig. 1 | Magnetic actuation system for untethered small robots equipped with a permanent magnet. a** Schematic of a magnetic actuation system for untethered capsule robots for examination of gastrointestinal tracts. A multi-DoF serial manipulator can be used to move the actuating magnet to control the magnetic field and field gradient at the location of the magnet on the robot. The magnetic object onboard the robot can be either a permanent magnet or a soft magnet as in the proposed system in this paper. **b** Schematic of magnetic torque on a permanent magnet. The magnetic torque is always located on the plane perpendicular to the magnetic moment vector of the magnet, resulting in zero torque component and inability of orientation manipulation about the axis of magnetic moment. **c**, **d** Schematic of magnetic force on a permanent magnet orientated along **y**- and **x** axis, respectively. For example, the magnetic gradient of $\frac{\partial \mathbf{B}}{\partial x} = [1,0]^{\mathrm{T}}$ T/m and $\frac{\partial \mathbf{B}}{\partial y} = [0,1]^{\mathrm{T}}$ T/m leads to magnetic force $\mathbf{f} = [0,1]^{\mathrm{T}}$ N and $\mathbf{f} = [1,0]^{\mathrm{T}}$ N when the magnet (magnetic moment $\mathbf{m} = 1$ A·m²) is orientated along **y** and **x**-axis, respectively.

magnetic materials and soft-magnetic materials, depending on the different magnetic behaviors under magnetization of the applied field. It should be noted that magnetization **M** is also used as a characteristic quantity to describe the magnetic strength and orientation of a magnetic object under magnetization, and it is related to the magnetic moment **m** by

$$\mathbf{m} = \nu \mathbf{M} \qquad (3)$$

Where $\nu$ is the volume of the magnetic object. Hard-magnetic materials have large hysteresis and coercivity (the extra magnetic field to bring the magnetization back to zero) under magnetization. At the extreme of hard-magnetic materials are permanent magnets, whose magnetization remains nearly constant for a large range of applied fields. Conventional magnetic actuation systems for small or micro robotic manipulation normally utilize a permanent magnet onboard the robot. On the other hand, soft magnets have very small coercivity and negligible hysteresis and show a linear magnetization up to the point of saturation. In the context of magnetic robotics, we assume a moderate applied magnetic field so the linear magnetization-field relation always holds for soft magnets.

The magnetization **M** of a soft magnet is related to the internal magnetic field $\mathbf{H_{in}}$ by magnetic susceptibility $\chi$:

$$\mathbf{M} = \chi \mathbf{H_{in}} \qquad (4)$$

where $\chi$ is related with magnetic permeability $\mu$ by $\mu = \mu_0(1+\chi)$. $\mu_0$ is magnetic permeability in vacuum. The internal magnetic field $\mathbf{H_{in}}$ is the sum of the applied external magnetic field **H** and a resulting demagnetizing magnetic field $\mathbf{H_d}$, which is in the opposite direction of **H** and proportional to the magnetization **M**:

$$\begin{aligned} \mathbf{H_{in}} &= \mathbf{H} + \mathbf{H_d} \\ &= \mathbf{H} - \mathbf{NM} \end{aligned} \qquad (5)$$

where **N** is the demagnetization factor tensor. It is noted that both magnetic field strength **H** and magnetic flux density **B** can be used to describe the magnetic field and they are related by $\mathbf{B} = \mu_0 \mathbf{H}$ in vacuum.

The demagnetization tensor becomes a diagonal matrix if the coordinate frame is aligned with the principal axes of the soft magnet:

$$\mathbf{N} = \mathrm{diag}(n_a, n_b, n_c) \tag{6}$$

The demagnetization factors $n_a$, $n_b$ and $n_c$ depend on the geometry of the soft magnet[52–54] (Supplementary Note 1). The demagnetization factor is between 0 and 1 and it tends to be smaller as the geometry in that direction becomes longer. The sum of three demagnetization factors is always unity ($n_a + n_b + n_c = 1$). Combining (4) and (5), it is obtained that

$$\mathbf{M} = \begin{bmatrix} \frac{\chi}{1+\chi n_a} & 0 & 0 \\ 0 & \frac{\chi}{1+\chi n_b} & 0 \\ 0 & 0 & \frac{\chi}{1+\chi n_c} \end{bmatrix} \mathbf{H} \approx \begin{bmatrix} \frac{1}{n_a} & 0 & 0 \\ 0 & \frac{1}{n_b} & 0 \\ 0 & 0 & \frac{1}{n_c} \end{bmatrix} \mathbf{H} \tag{7}$$

where the approximation in the final step is reasonable for soft-magnetic materials with $\chi \gg 1$. It is found from (7) that the magnetization in each axis is obtained by scaling the applied magnetic field by the inverse of the corresponding demagnetization factor in that axis (Fig. 2a). The magnetization is anisotropic for a non-spherical soft magnet with unequal demagnetization factors in different axes. If the demagnetizing factors are different in different axes, the directions of **M** and **H** become different, resulting in a non-zero magnetic torque on the soft magnet (Fig. 2b). It is also shown that both the magnitude and orientation of the magnetization of a soft magnet varies with the applied field, which is fundamentally different from permanent magnets with a fixed magnetization. Fig. 2d shows the variation of magnetization with the direction of an applied magnetic field in the central plane of an ellipsoidal soft magnet. The magnetization is aligned with the applied field as it is pointed along the long axis. As the applied field rotates towards the short axis, the magnetization vector lags behind the applied field but catches up to align with the applied field again as it arrives at the short axis. Since the applied field can be controlled to point to any direction in the general three-dimensional case, the resulting magnetization and magnetic torque can also be pointed to any direction (Fig. 2b), which opens up the possibility of 3-DoF torque generation and orientation manipulation of a soft magnet.

Since both the magnetization and the resulting magnetic torque depends on geometry-related demagnetization factors, it is necessary to analyze the three-dimensional magnetization behavior of soft magnets with different geometries and find feasible geometries for 3-DoF orientation manipulation. Sphere has equal dimensions ($a = b = c$) and thus equal demagnetization factors in three principal axes, resulting in isotropic magnetization and constant alignment of the magnetization vector with the applied field (Fig. 2e). Therefore, the magnetic torque on the spherical soft magnet is always zero. For the prolate ellipsoid ($a > b = c$), the magnetization is aligned with the applied field when it is pointed along the unequal **a**-axis ($\mathbf{H_1}$ in Fig. 2f) and to any direction within **boc**-plane ($\mathbf{H_2}$ in Fig. 2f), resulting in zero torque in these cases. When the magnetic field is applied in other arbitrary directions ($\mathbf{H_3}$ in Fig. 2f), **M, H**, and the **a**-axis are co-planar due to the rotational symmetry about the **a**-axis, resulting in a magnetic torque always within **boc**-plane. Therefore, similar to the case of permanent magnets, no torque can be generated about the **a**-axis and only 2-DoF orientation manipulation is achievable. Similar phenomenon exists for the oblate ellipsoid ($a = b > c$), no torque can be generated about the unequal **c**-axis (Fig. 2g). Therefore, it is necessary to make sure that the geometry of the soft magnet is anisotropic in all three principal directions to achieve 3-DoF torque generation. For the anisotropic tri-axial ellipsoid ($a > b > c$), when the magnetic field is applied within **aob**-plane ($\mathbf{H_1}$ in Fig. 2h), **boc**-plane ($\mathbf{H_2}$ in Fig. 2h), and **aoc**-plane ($\mathbf{H_3}$ in Fig. 2h), the magnetic torque can be generated about

**c**-axis, **a**-axis, and **b**-axis, respectively. The torque in any direction can be generated by tuning the direction of the applied magnetic field ($\mathbf{H_4}$ in Fig. 2h). Besides tri-axial ellipsoids, anisotropic elliptical cylinders ($a \neq b \neq h$) and cuboids ($w \neq l \neq h$) are also feasible geometries of the soft magnet for 3-DoF torque generation and orientation manipulation (Fig. 2c). In this paper, anisotropic elliptical cylinders are used in experiments while theoretical analysis and schematic illustration are based on tri-axial ellipsoids. In practice, elliptical cylinders or cuboids are recommended for their simplicity of machining. Geometric dimensions of the soft magnet can be designed to obtain the desired demagnetization factors that meet the needs of robot manipulation (Supplementary Note 1). The behavior of magnetization and orientation manipulation does not depend on the shape of the soft magnets as long as their demagnetization factors are the same.

According to Eq. (7), The variation of magnitude and orientation of magnetization **M** with the direction of a constant-magnitude applied field **H** can be quantitatively investigated. Denote the orientation of **H** relative to the soft magnet by elevation $\varphi$ and azimuth $\theta$. Denote the orientation of the resulting **M** by elevation $\beta$ and azimuth $\alpha$. Then the following relations can be obtained.

$$\alpha = \arctan\left(\frac{n_b}{n_c}\tan\theta\right)$$
$$\beta = \arctan\left(\sqrt{\frac{n_b^2 n_c^2 \tan^2\varphi}{n_a^2 n_c^2 \cos^2\theta + n_a^2 n_b^2 \sin^2\theta}}\right) \tag{8}$$
$$|\mathbf{M}| = \frac{\sin\varphi}{n_\alpha \, \sin\beta}|\mathbf{H}|$$

The above relations are depicted in Fig. 2i. The azimuth of the magnetization vector $\alpha$ is only a function of the azimuth of the applied field $\theta$ and demagnetization factors $n_b$ and $n_c$, with no dependence on the elevation $\varphi$. As the ratio of $n_b$ to $n_c$ becomes smaller (the geometry becomes more anisotropic in the direction of **b**-axis and **c**-axis), the relation between $\alpha$ and $\theta$ becomes more nonlinear, creating a larger magnetization lag in the azimuthal angle. The elevation and magnitude of the magnetization vector depends on both the elevation and the azimuth of the applied field (Fig. 2i). It is also known from Eq. (8) that $\beta$ and $\|\mathbf{M}\|$ depends on relative magnitude of demagnetization factors and thus on relative geometric dimensions in three principal axes of the soft magnet. In summary, the anisotropic geometry of the soft magnet causes its anisotropic magnetization in the applied field, which further leads to misalignment of the magnetization vector and the applied field and thus enables generation of 3-DoF magnetic torques.

## Magnetic torque on the soft magnet

From the qualitative analysis in the last section, it is known that 3-DoF torques can be generated on the soft magnet about each of its principal axis due to anisotropic magnetization. In this section, the 3-DoF magnetic torques on the soft magnet are quantitatively investigated. It is known from Eq. (2) that the magnetic torque can be obtained by taking the cross product of the magnetic moment of the soft magnet and the externally applied field. Substituting Eq. (7) into (2) gives the magnetic torque about each principal axis of the anisotropic soft magnet:

$$\tau_a = \mu_0 \upsilon \frac{|n_c - n_b|}{2 n_b n_c} |\mathbf{H}|^2 \, \sin(2\theta)\cos^2(\varphi)$$
$$\tau_b = \mu_0 \upsilon \frac{|n_c - n_a|}{2 n_a n_c} |\mathbf{H}|^2 \, \sin(\theta)\sin(2\varphi) \tag{9}$$
$$\tau_c = \mu_0 \upsilon \frac{|n_b - n_a|}{2 n_b n_a} |\mathbf{H}|^2 \, \cos(\theta)\sin(2\varphi)$$

where $\tau_a$, $\tau_b$, $\tau_c$ are magnetic torques about **a**-axis, **b**-axis, and **c**-axis. The surface plots of magnetic torques with the direction of the applied

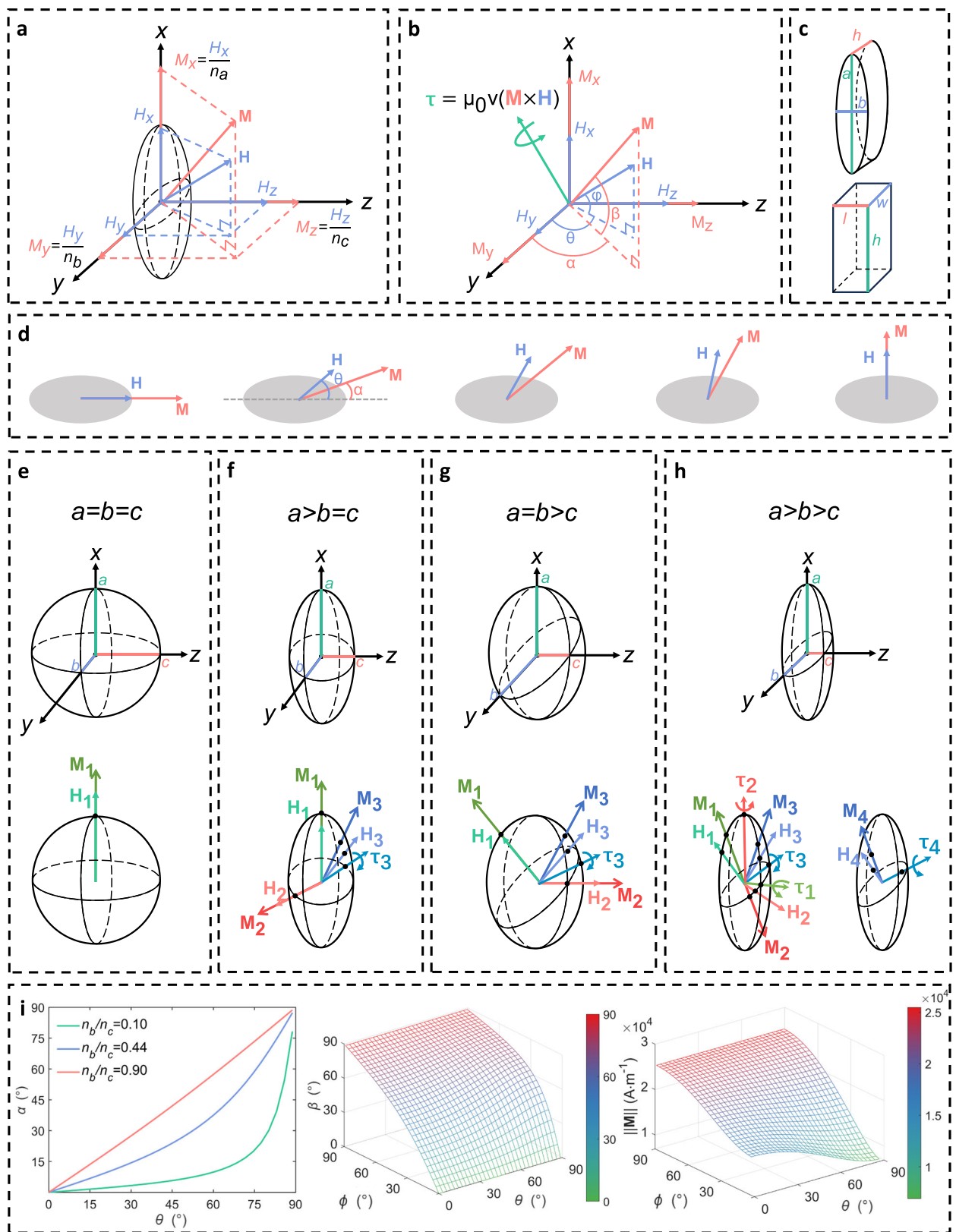

field are shown in Fig. 3a, which again validates that non-zero torque can be generated about all three principal axes. The magnitude and direction of the magnetic torque vector can be controlled by adjusting the orientation of the applied field even if the magnitude of the applied field is kept constant. A blue curve is drawn on each of the surface plots of magnetic torques, representing the variation of magnetic torque when the external magnetic field is applied within **boc**-plane,

**coa**-plane, and **aob**-plane. For example, when the applied field with constant magnitude varies its orientation within **boc**-plane, a magnetic torque about **a**-axis is generated and it always tends to align the longer **b**-axis (either the positive or the negative end depending on whichever is closer to the applied field) of the cross-sectional ellipse with the applied field. It is noted that the maximum torque about each axis exists on these curves and it is achieved when the angle between the

**Fig. 2 | Magnetization of a soft magnet with an anisotropic geometry.**
**a** Schematic of anisotropic magnetization of an ellipsoidal soft magnet. The mag-
netization component in three principal axes is proportional to the applied field
component by a scaling factor of the inverse of the demagnetization factor.
**b** Schematic of magnetic torque on the ellipsoidal soft magnet shown in **a**. The
anisotropic magnetization of the soft magnet leads to separation of the direction of
magnetization vector **M** from the applied field **H**, resulting in a non-zero magnetic
torque $\tau = \mu_0 v(\mathbf{MH})$. The orientation of **H** is denoted by elevation $\varphi$ and azimuth $\theta$.
The orientation of **M** is denoted by elevation $\beta$ and azimuth $\alpha$. **c** Alternative geo-
metries of soft magnets for anisotropic magnetization: Cuboids with $w \neq l \neq h$ and
elliptical cylinders with $a \neq b \neq h$. **d** The variation of magnetization vector with the
orientation of the applied field in the central plane of the ellipsoidal soft magnet. As
the applied field rotates from the major axis to the minor axis, the lag angle
between the magnetization vector and the applied field first increases from zero

and then reduces to zero again. **e** Magnetization of a spherical soft magnet. The
magnetization vector is always aligned with the applied field with no torque gen-
erated. **f** Magnetization of a prolate ellipsoidal soft magnet. **M**, **H**, and the longest **a**-
axis are always co-planar, resulting in a magnetic torque always located in **boc**-
plane with zero torque component in **a**-axis. **g** Magnetization of an oblate ellip-
soidal soft magnet. **M**, **H**, and the shortest **c**-axis are always co-planar, resulting in a
magnetic torque always located in **aob**-plane with zero torque component in **c**-axis.
**h** Magnetization of a triaxially ellipsoidal soft magnet. Magnetic torque can be
generated in all directions by tuning the direction of the applied field. **i** The var-
iation of azimuth angle, elevation angle, and magnitude of magnetization **M** of a tri-
axial ellipsoidal soft magnet ($n_a = 0.16$, $n_b = 0.26$, $n_c = 0.58$, $\frac{n_b}{n_c} = 0.44$) with the
direction of the applied field. The variation of azimuth of **M** is also illustrated for
soft magnets with $\frac{n_b}{n_c} = 0.1$ and $\frac{n_b}{n_c} = 0.9$.

applied field and the major axis of the cross-sectional ellipse is
45° (Fig. 3d).

The maximum torque about one axis characterizes the capability
of orientation actuation about that axis. When the maximum torque is
achieved, the trigonometric part of the torque expressions in (9) takes
the value of 1, giving the following identity.

$$\tau_{b,\max} = \tau_{a,\max} + \tau_{c,\max} \qquad (10)$$

where it is assumed that $a > b > c$ and $n_a < n_b < n_c$. Note that the max-
imum torques about three principal axes cannot be achieved simul-
taneously. Eq. (10) shows that the largest maximum torque is about **b**-
axis (second-longest axis) and it is the sum of maximum torques about
the remaining two axes. It is found that the capability of torque gen-
eration is unequal about different axes in the soft-magnet actuation
system, which is quite different from the case of traditional
permanent-magnet system that has equal torque generation capability
about two axes (permanent-magnet system cannot generate torque
about the third axis of magnetic moment). Experiments are conducted
to validate the theoretical model of magnetic torques on an
anisotropic soft magnet. An orthogonal Helmholtz coil system is used
to generate a uniform magnetic field in its inner workspace for
orientation manipulation of the soft-magnetic robot (Supplementary
Figs. 1 and 2). The orientation and magnitude of the applied field can be
adjusted by tuning the current of each pair of coils. In this experiment
of torque validation, the capsule robot with a soft magnet is suspended
in the water and placed in the workspace of the Helmholtz coils. The
soft magnet is first oriented with an angle of 45° relative to the applied
field and then released to rotate until aligning with the applied field.
The maximum magnetic torques can be indirectly computed from the
rotation time based on dynamic equations with varying angular
accelerations (See the section of Methods). The experimental values of
the maximum torques about principal axes agree well with the
theoretical values for different magnitudes of the applied field (Fig. 3b).
The identity (10) is also validated, as shown in Fig. 3c. Therefore, the
above modeling of the anisotropic magnetization of the soft magnet
and the resulting torque generation is valid.

### Orientation stability of an anisotropic soft magnet in a uniform magnetic field

In permanent-magnet-based actuation systems, the magnetic torque
always tends to align the axis of magnetic moment of the permanent
magnet with the applied field. If the permanent magnet is controlled
with moderate dynamics and the orientation of the applied field varies
slowly, the orientation lag of the permanent magnet relative to the
applied field can be ignored and it can be assumed that the orientation
of the magnet (2-DoF direction of the magnetic moment) varies with
the direction of the applied field synchronously. This stabilizing torque
enables open-loop 2-DoF orientation manipulation of a permanent

magnet by controlling only the direction of applied field without the
need of orientation feedback.

As for the anisotropic soft magnet, there are three principal axes
and its 3-DoF orientation stability in an applied field becomes more
complicated than the permanent magnet case. To investigate the
orientation stability of an anisotropic soft magnet, it is embedded in a
capsule shell that is submerged in water and placed inside the work-
space of the orthogonal Helmholtz coils as shown in Supplementary
Fig. 2, where a constant magnetic field is applied. Fig. 4 shows the
experimental results of orientation stability of an anisotropic soft
magnet under perturbation about different principal axes. For each
case, both photographic and schematic illustrations of the soft-magnet
robot in each state are provided. In Fig. 4a, the **a**-axis (longest) of the
soft magnet is aligned with the applied field initially, generating zero
magnetic torque. Then the soft magnet is perturbed about **a**-axis, **b**-
axis, and **c**-axis for positive 45° (right-hand rule), respectively. Finally,
the soft magnet is released and its behavior is observed (Supplemen-
tary Movie 1). Since the magnetic torque is always zero as long as the
principal axes are aligned with the applied field, the soft magnet will
stay where it is perturbed about **a**-axis after release. This state is
defined as marginally stable. When the soft magnet is released after
perturbation about **b**- and **c**-axis, it will recover back to the initial state
under the stabilizing torque, i.e. aligned with the applied field. This
state is defined as stable. Fig. 4b shows the stability of the soft magnet
when its **b**-axis (second-longest) axis is aligned with the applied field in
the initial state (Supplementary Movie 2). When it is perturbed about **a**-
axis, **b**-axis, and **c**-axis, the dynamic behaviors are stable, marginally
stable, and unstable, respectively. In the unstable state, the soft mag-
net continues to rotate away from the initial state after release until it
reaches the next stable state. Fig. 4c shows the stability of the soft
magnet when its **c**-axis (shortest axis) is aligned with the applied field in
the initial state (Supplementary Movie 3). When it is perturbed about **a**-
axis, **b**-axis, and **c**-axis, the dynamic behaviors are unstable, unstable,
and marginally stable, respectively. Therefore, the longest axis of an
anisotropic soft magnet is the most stable axis, with the strongest
tendency to align with the applied field. The shortest axis is the most
unstable axis, with the weakest tendency to align with the applied field.

When the magnetic field is applied in an arbitrary direction (not
aligned with principal axes), the soft magnet always tends to align its
longest **a**-axis with the magnetic field. In the process of long-axis
alignment, the second longest **b**-axis tends to align with the projection
of the applied field on the **boc**-plane under the torque about **a**-axis.
However, complete **b**-axis alignment is not guaranteed since the tor-
que about **a**-axis becomes zero once the **a**-axis aligns with the applied
field. The longest axis of an anisotropic soft magnet is analogous to the
axis of magnetic moment of a permanent magnet. By controlling the
direction of a single applied magnetic field, only 2-DoF orientation of
the soft magnet can be manipulated with the longest axis aligning with
the applied field. The remaining degree of freedom, i.e., the orientation
about **a**-axis, cannot be deterministically and precisely controlled by a

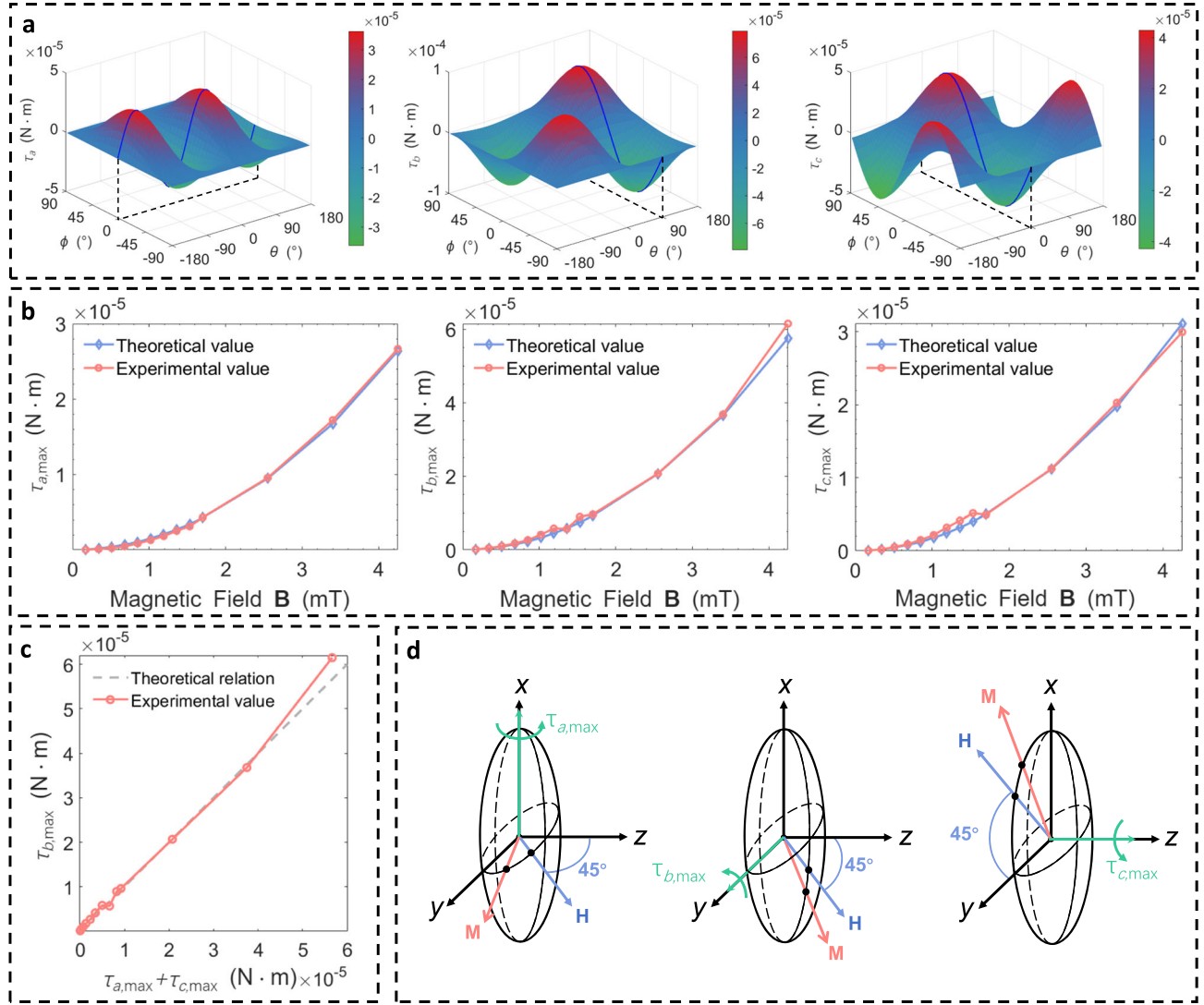

**Fig. 3 | Magnetic torque on an anisotropic soft magnet. a** The variation of magnetic torques about **a**-axis, **b**-axis, and **c**-axis with the direction of the applied magnetic field. The blue curves denote the magnetic torque when the magnetic field is applied within **boc**-plane, **coa**-plane, and **aob**-plane. **b** Comparison of experimental value and theoretical value of maximum torques about **a**-, **b**-, and **c**-axis under different magnitudes of the applied magnetic field, respectively. **c** Validation of the identity relation of maximum torques about **a**-, **b**-, and **c**-axis, i.e. $\tau_{b,\max} = \tau_{a,\max} + \tau_{c,\max}$. **d** Schematic of maximum magnetic torques about **a**-, **b**-, and **c**-axis when the magnetic field is applied 45° from the major axis of the cross-sectional ellipse in **boc**-plane, **coa**-plane, and **aob**-plane.

single applied field although it can be adjusted by a varying magnetic torque about **a**-axis. This fact suggests that precise 3-DoF orientation manipulation for an anisotropic soft magnet is not necessarily achievable even if magnetic torques about three principal axes can be generated.

### Alternating and open-loop control of 3-DoF orientation of the soft-magnet robot

Based on the analysis of magnetic torque generation and orientation stability of an anisotropic soft magnet, it is found that there are two major challenges for precise 3-DoF orientation control of the soft magnet:

(1) The rotation about the longest **a**-axis cannot be simultaneously controlled in an open-loop manner together with the other two DoF (aligning **a**-axis with the applied field) using a single actuating magnetic field.

(2) The more stable axes (**a**-axis or **b**-axis) of the anisotropic soft magnet could be orientated reversely from the desired direction by simply setting the actuating magnetic field in that direction.

The first problem is already discussed in the previous section on orientation stability. The second problem arises from the fact that the magnetic torque can possibly align either the positive or the negative longer axis of the soft magnet with the applied field, depending on whichever end is closer to the applied field. In other words, if the angle between the positive longer axis and the applied field is less than 90°, then the soft magnet aligns its positive axis with the applied field. Otherwise, the negative axis aligns with the applied field. This behavior is fundamentally different from that of a permanent magnet in magnetic actuation, where the permanent magnet always tends to orient its north pole along the applied field. To address these two challenges regarding magnetic actuation of a soft magnet, an alternating and open-loop control strategy is proposed to control the direction of the longest axis (2 DoFs) and the orientation about the longest axis (1-DoF) separately over small-angle steps using two alternatingly applied actuating magnetic fields (Fig. 5). The 3-DoF orientation of the soft magnet can be manipulated to reach any arbitrary target orientation from an initial orientation. First, the orientation path between the initial orientation and the target orientation is discretized using the

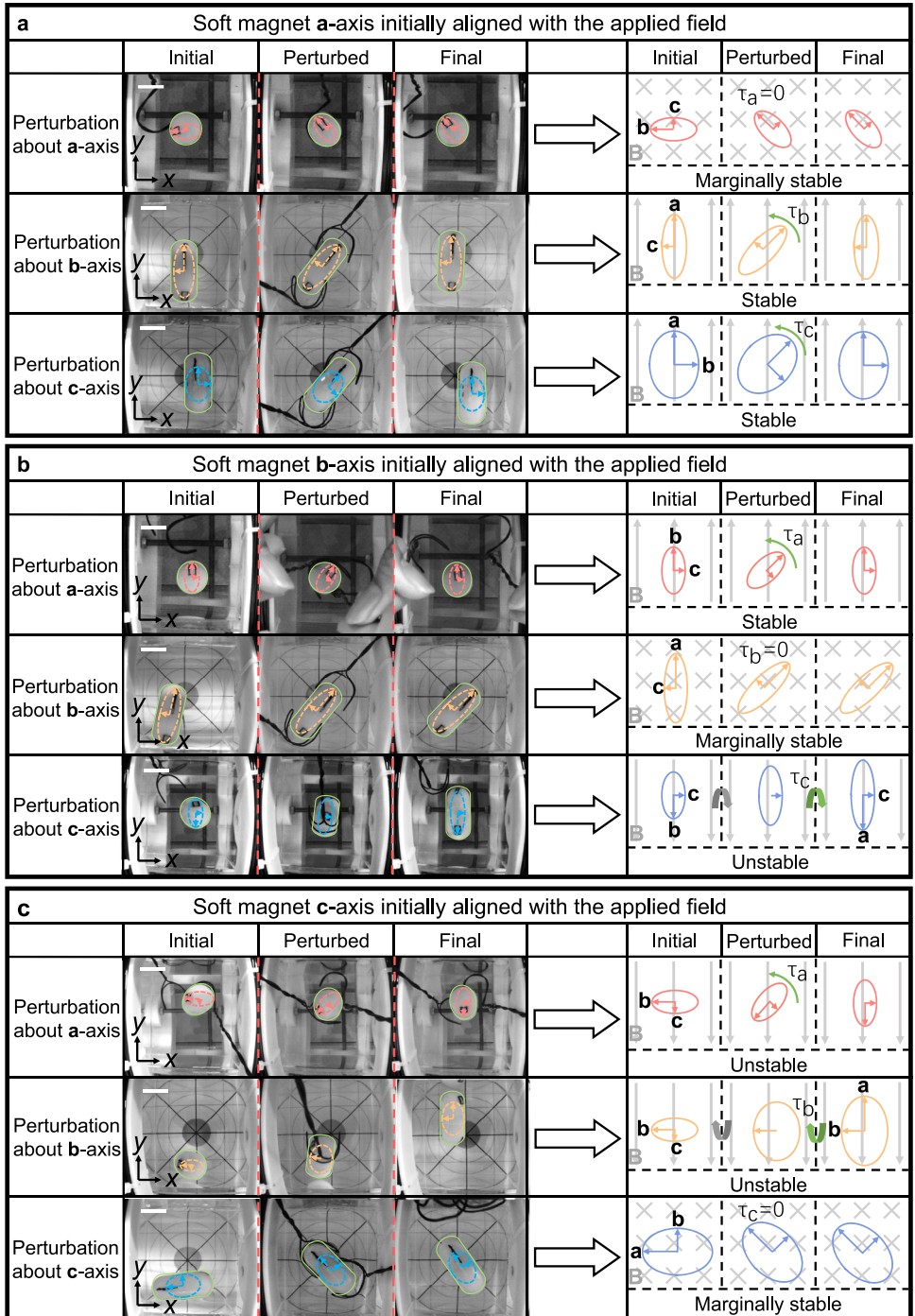

**Fig. 4 | Orientation stability of a small soft-magnet robot suspended in water.**
**a** Orientation stability about three principal axes under perturbations when **a**-axis of the soft magnet is aligned with the applied field. The soft magnet is marginally stable about **a**-axis and stable about **b**- and **c**-axis. **b** Orientation stability about three principal axes under perturbations when **b**-axis of the soft magnet is aligned with the applied field. The soft magnet is stable about **a**-axis, marginally stable about **b**-axis, and unstable about **c**-axis. **c** Orientation stability about three principal axes under perturbations when **c**-axis of the soft magnet is aligned with the applied field. The soft magnet is unstable about **a**- and **b**-axis, and marginally stable about **c**-axis. In each case, the soft-magnet robot is perturbed for positive 45° (right-hand rule). In photographs, solid outlines denote the capsule robot while dashed outlines denote the ellipsoidal soft magnet inside. In schematics, only the soft magnet is illustrated with solid outlines. The gray arrows in schematics indicate the applied magnetic field. The scale bars indicate 2 cm.

angle-axis interpolation that gives the shortest path. The discretization step is chosen sufficiently small such that the relative angle between two adjacent midway orientations (green coordinate frames in Fig. 5) is smaller than 90° to prevent problem 2 above. Then the actuating magnetic field $\mathbf{B}_i^a$ that controls the direction of the longest axis is applied along **a**-axis of the next midway orientation such that a magnetic torque is generated to align the longest axis along the desired **a**-axis of the midway orientation. Next, the actuating magnetic field $\mathbf{B}_i^r$ that controls the rotation about the longest axis is applied along **b**-axis of the next midway orientation such that the second-longest axis of the soft magnet aligns with the desired **b**-axis of the midway orientation. By implementing this two-step control strategy alternately, full 3-DoF orientation of the soft magnet can be precisely controlled to reach each midway orientation and finally reach the desired target

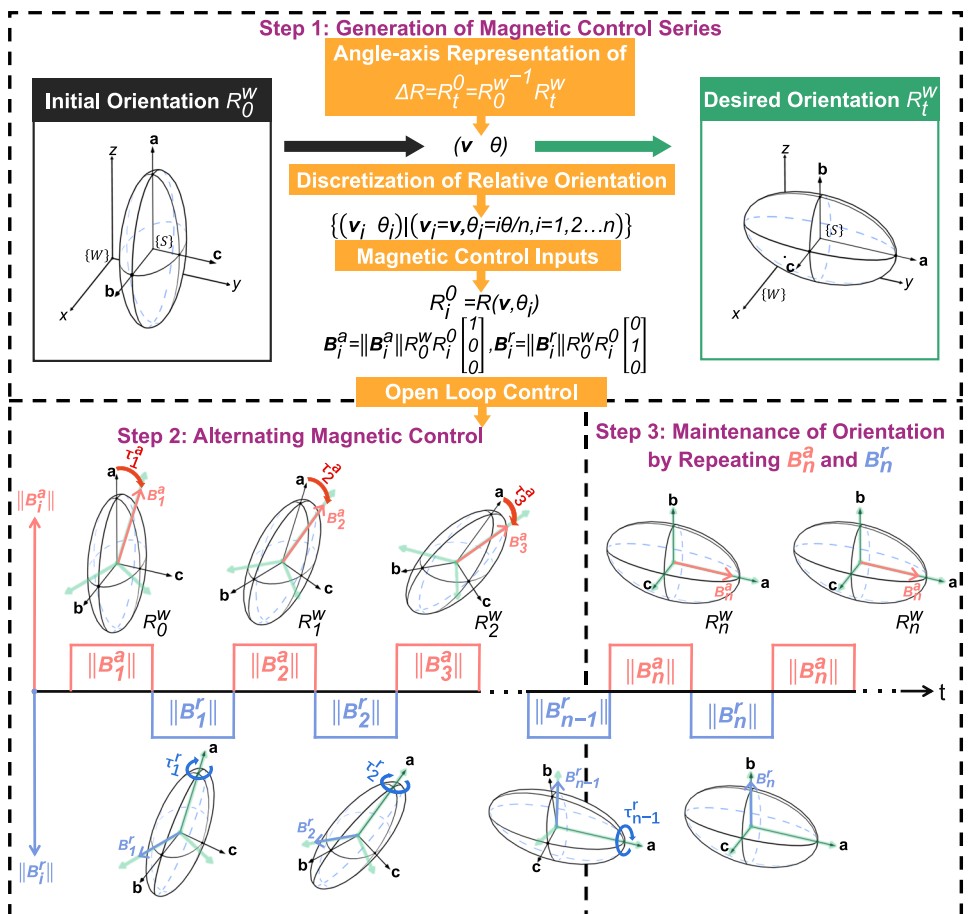

**Fig. 5 | Alternating open-loop control strategy for 3-DoF orientation manipulation of an anisotropic soft magnet.** In the first step, the orientation path is discretized into small orientation steps and orthogonal magnetic control inputs $\mathbf{B}_i^a$ and $\mathbf{B}_i^r$ are generated along **a**-axis and **b**-axis of the discretized midway orientations of the anisotropic soft magnet. In the second step, the orthogonal magnetic control inputs $\mathbf{B}_i^a$ and $\mathbf{B}_i^r$ are applied alternatingly to control the direction of the longest axis (2 DoFs) and the rotation about the longest axis (1 DoF), respectively, in each discrete orientation step until the soft magnet reaches the target orientation. The green coordinate frames denote the next desired midway orientation. In the last step, alternating orthogonal magnetic fields need to be maintained to "lock" the soft magnet to the desired orientation for disturbance rejection.

orientation. Since the orientation adjustment during each discrete step is very small and the dynamic process ends very quickly (<7 ms), the proposed alternating orientation control strategy can be regarded as "locking" the longest (most stable) axis and second-longest (second most stable) axis of the anisotropic soft magnet using two orthogonal actuating magnetic fields and drag the soft magnet towards the desired target orientation over multiple small steps.

If the soft magnet is required to precisely stay at the target orientation, the two actuating magnetic fields should be still alternatingly maintained along its longest and second-longest axis to "lock" the soft magnet for disturbance rejection. Another issue with the open-loop control strategy is to determine the initial orientation of the soft magnet. When we apply two alternating magnetic fields $\mathbf{B}_0^a$ and $\mathbf{B}_0^r$ to align the magnet with the desired initial orientation, the actual orientation of the soft magnet might be flipped due to the problem 2 above. The true resulting orientation is one of the four possible orientations (Supplementary Fig. 3). Therefore, external observation (e.g., a camera, a low-accuracy orientation sensor, or human observation, etc.) is needed to determine the true initial orientation from four possibilities.

The open-loop 3-DoF orientation manipulation of the soft-magnetic robot is experimentally demonstrated using the proposed alternating control strategy. In the first experiment, the capsule robot is rotated in a single degree of freedom about **a**-axis, **b**-axis, and **c**-axis of the embedded anisotropic soft magnet for 90°, respectively (Fig. 6a, Supplementary Movies 4–6), which shows that all three DoFs of

orientation can be precisely manipulated using the soft-magnet based actuation method. Note that the alternating magnetic control is necessary even for precise 1-DoF rotation since the control method using a single magnetic field still suffers from the problem of orientation instability under potential disturbances. In the second experiment, the capsule robot is first rotated about **x**-axis of the global coordinate frame for 45°, then rotated about global **y**-axis for 180°, then rotated about global **z**-axis for 45°, and finally rotated about **k**-axis ($\hat{\mathbf{k}} = [-0.28, -0.68, 0.68]^T$) for 63° to arrive at the final orientation (Fig. 6b, Supplementary Movie 7). The top view (**xoy**-plane projection) and the front view (**yoz**-plane projection) of the manipulated robot are provided, with at least one view validating the accuracy of orientation control by showing the special orientation angle (e.g., 45°) of the projected robot. This experiment shows that the proposed alternating control strategy can achieve full 3-DoF dexterous manipulation of a soft-magnet capsule robot through any complicated orientation path. Three components of the alternating magnetic control inputs (i.e., voltage inputs to orthogonal Helmholtz coils) along the orientation path are also shown in Fig. 6c. The orthogonal magnetic fields for alignment of the longest axis and rotation about the longest axis are switched and their direction is varying according to the planned orientation trajectory to drag the soft magnet to change orientation synchronously. The alternating frequency of the orthogonal magnetic fields is 10 Hz in Fig. 6. The upper limit of the frequency of the alternating magnetic control is around 10 kHz due to the inductance effect

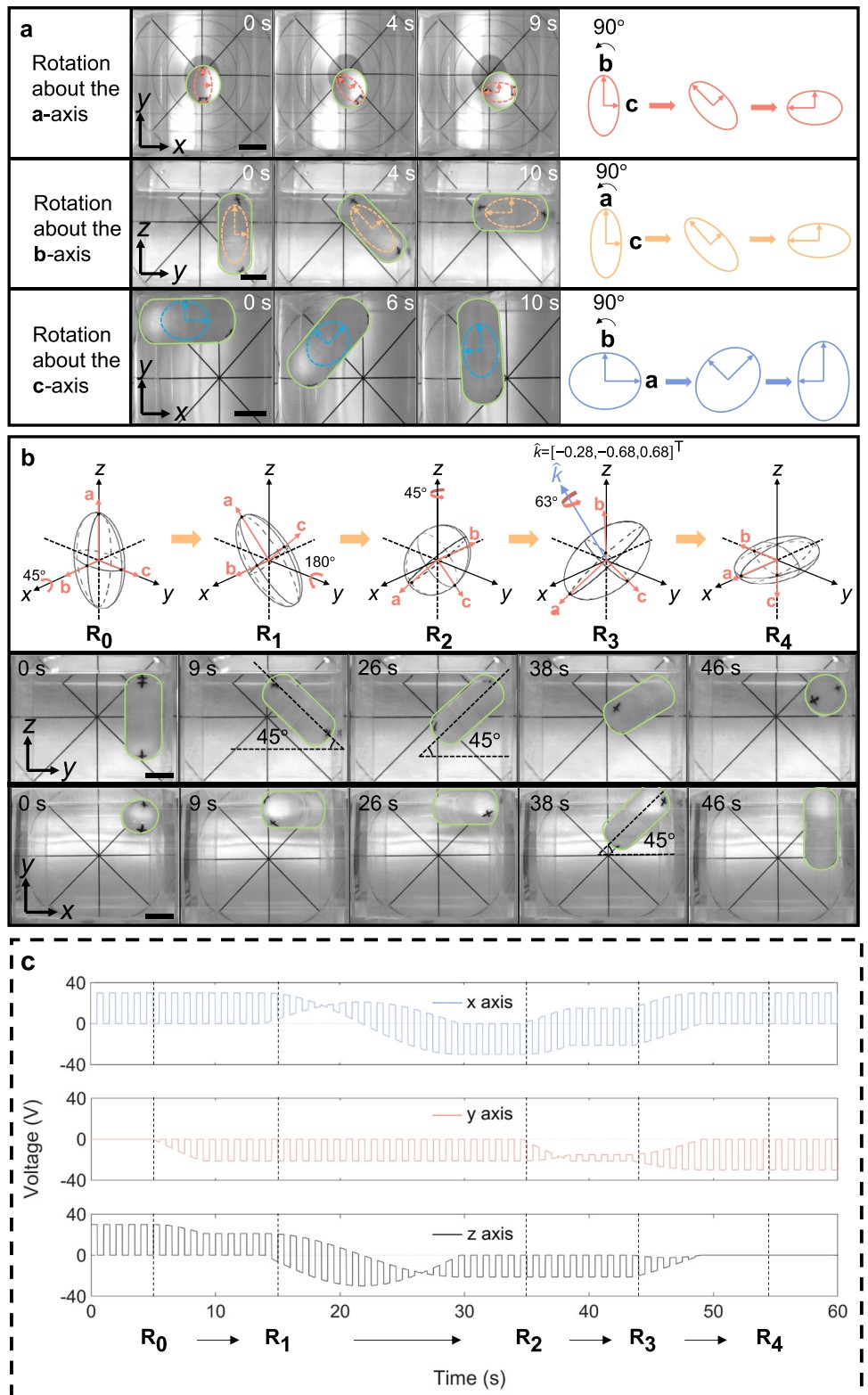

**Fig. 6 | Experimental demonstration of open-loop 3-DoF orientation manipulation of the soft-magnet-based robot. a** Sequence of photos and schematics of robot rotation about **a**-, **b**-, and **c**-axis of an anisotropic soft magnet for 90°. **b** Sequence of schematics and photos that demonstrate the 3-DoF orientation manipulation of the soft-magnet-based robot over an arbitrary orientation path. The capsule robot is first rotated about global **x**-axis for 45°, then rotated about global **y**-axis for 180°, then rotated about global **z**-axis for 45°, and finally rotated about **k**-axis for 63° to arrive at the final orientation. A top view (**xoy**-plane projection) and a front view (**yoz**-plane projection) of the manipulated robot are provided, with at least one view validating the accuracy of orientation control by showing the special orientation angle (e.g., 45°) of the projected robot. **c** Voltage inputs to three orthogonal Helmholtz coils for controlling the magnetic field components in three dimensions. The voltage inputs are alternated to generate alternating magnetic control inputs $\mathbf{B}_i^a$ and $\mathbf{B}_i^r$. The frequency of the alternating magnetic fields is 10 Hz (The plot shows a lower frequency for better demonstration). The scale bars indicate 2 cm.

of the Helmholtz coils (Supplementary Movie 10). On the other hand, too low a magnetic control frequency causes non-smooth orientation manipulation and failure of stabilization at the target orientation. The lower limits of the frequency of the alternating magnetic control for smooth and stable orientation manipulation is around 2 Hz and 0.5 Hz, respectively (Supplementary Movie 9).

Additional experiments are conducted with smaller capsule robots down to the size of commercial capsule endoscopes ($\varnothing 14$ mm × 28 mm) and smaller soft magnets down to the size of $10 \times 1.2 \times 4.8$ mm (Supplementary Fig. 4). Similar accuracy of 3-DoF orientation control is achieved, which validates the applicability of the proposed soft-magnet actuation method. In addition, the soft-magnet-based orientation manipulation is also experimentally evaluated in the presence of the flow disturbance using a water circulation system (Supplementary Figs. 5 and 6 and Supplementary Movie 8). It is shown that the introduced flow disturbance has a negligible impact on 3-DoF orientation control of the robot, which further validates the robustness of the proposed soft-magnet actuation method.

## Discussion

This paper proposes a new magnetic actuation principle that exploits another type of magnetic materials, i.e., soft magnets, for magnetic manipulation. Designed with an anisotropic geometry such as a tri-axial ellipsoid, the soft magnet exhibits different magnitudes of magnetization along its three principal axes in the presence of an external magnetic field. The anisotropic magnetization of the soft magnet leads to separation of magnetization vector (or magnetic moment) from the applied field, resulting in magnetic torque generation. Theoretical modeling and experiments show that magnetic torques can be generated about all three principal axes of the anisotropic soft magnet, which brings the capability of 3-DoF orientation manipulation. However, 3-DoF orientation manipulation cannot be simply achieved by controlling a single actuating magnetic field in an open-loop manner since the orientation stability of three principal axes of the soft magnet is different from one another. Therefore, we propose an alternating open-loop control strategy, where a pair of orthogonal actuating magnetic fields are applied alternatingly at small steps to precisely control the direction of the longest axis of the soft magnet and the rotation about the longest axis, respectively. Compared with other 3-DoF orientation manipulation methods based on permanent magnets, the proposed method relies on a single magnetic body with a simple geometry (e.g., elliptical cylinder) that is easy to fabricate and requires no complicated assembly or patterned magnetization.

Although the soft-magnet-based magnetic actuation method is fundamentally investigated and demonstrated for 3-DoF orientation manipulation of small robots, there are still some limitations and open challenges left for future work. First, the magnetic torque generated on a soft magnet is much smaller than that on a permanent magnet of similar size. For example, if an external magnetic field of $5 \times 10^{-3}$ T is applied, the maximum torque on the soft magnet used in experiments of this paper is $6 \times 10^{-5}$ N·m while the maximum torque on a permanent magnet of similar size (1695 mm³) with a magnetic moment of 1.08 A·m² is $5.4 \times 10^{-3}$ N·m. If the small robot is manipulated in the liquid environment with negligible friction, then the small torque is sufficient to lock **a**-axis and **b**-axis of the soft magnet to the two actuating magnetic fields and thus the proposed orientation control strategy is expected to work well. Otherwise, stronger actuating magnetic fields might be required to ensure sufficient magnetic torque. However, an excessively strong magnetic field might saturate magnetization of the soft magnet, which induces more complicated dynamic behaviors and even impairs magnetic torque generation. We have assumed that the soft magnet is never saturated during magnetization throughout the paper. The 3-DoF rotational dynamics of soft-magnetic actuation under saturated magnetization remains to be investigated in future work.

The second limitation of this work is that only 3-DoF orientation manipulation is considered with the soft magnet placed in a uniform actuating magnetic field while 3-DoF translational manipulation is not investigated. Similar to the permanent-magnet-based actuation system, the magnetized soft magnet experiences 3-DoF magnetic force in the presence of magnetic field gradients (Supplementary Note 2). We have designed a preliminary control method for 6-DoF manipulation of the soft-magnet robot that decomposes orientation and translation control and builds upon the open-loop orientation control strategy proposed in this paper (Supplementary Note 2). Although it remains to be validated experimentally, the soft-magnet actuation method is a promising new method of 6-DoF manipulation of small-scale robots.

Finally, the capsule robot is manipulated in water with negligible friction in this work. In future work, the soft-magnet-based robot manipulation will be evaluated in more realistic scenarios such as in a gastrointestinal tract.

## Methods

### Soft magnet

The soft-magnetic material (Type No. 1J85, Dongbei Special Steel Group Co. Ltd.) used in experiments are mainly made of nickel (Ni), iron (Fe), and molybdenum (Mo) with a mass ratio of 80%, 15%, and 5%, respectively. It has a high maximum magnetic permeability of $\mu_m = 156.6$ mH/m and low coercivity of $H_c = 1.34$ A/m. The soft-magnetic materials are purchased as bars and then machined into elliptical cylinders, which can be embedded into a capsule robot for magnetic manipulation. The soft-magnetic cylinder has a weight of 14.8 g and a dimension of a semi-major axial length of 30 mm, a semi-minor axial length of 6 mm, and a height of 12 mm. The demagnetization factors are $n_a = 0.16$, $n_b = 0.26$, and $n_c = 0.58$, which corresponds to the semi-major axis, the axis of height, and the semi-minor axis, respectively.

### Small capsule robot

Since the small capsule robot in this work is used to validate and demonstrate the proposed soft-magnet-based actuation method rather than showing its real applications, the robot only consists of a plastic shell and an internal on-board soft magnet, with no endoscopic camera and other electronic components. The capsule shell is made of photosensitive engineering resin (UTR 8220) by high-precision stereolithography (SLA) 3D printing (Lite800HD, UnionTech3D Inc.). The shell is printed as two half pieces, which have an internal slot fixture to hold the soft magnet. Therefore, it is easy to assemble the capsule robot manually. The cylindrical capsule has a dimension of $\varnothing 25$ mm × 58 mm with a shell thickness of 1.2 mm. The size of the capsule robot is designed such that sufficient buoyancy can be generated to suspend the soft-magnet-embedded robot in the water.

### Tri-axial orthogonal helmholtz coils

For experimental validation of magnetic torque generation, orientation stability, and 3-DoF orientation manipulation, a magnetic actuation setup is built with tri-axial Helmholtz coils in orthogonal configuration (Supplementary Fig. 1). A Helmholtz coil is made of a pair of coaxial electromagnetic coils with equal radius, with the same current running with the same handedness. A nearly uniform magnetic field with negligible gradients can be generated along the axial direction inside the central region of the Helmholtz coil. Therefore, only magnetic torque can be generated on the soft magnet and there is no magnetic force. To generate omnidirectional uniform magnetic field, three pairs of Helmholtz coils are assembled orthogonally. The current supplies to three Helmholtz coils are tuned to control the magnitude and direction of the generated uniform field in the central workspace.

The current control system of the Helmholtz coils consists of three H-bridge boards (450 W DC motor driver module, Shaibang Inc.), three power sources (IT6302A, ITECH Co. Ltd.), and a microcontroller

(STM32F103C6T6, STMicroelectronics). Each of the H-bridge boards is powered by a 30 V power source and its output terminals are connected to one of the three Helmholtz coils. Magnetic control inputs are computed based on the manipulation task and then converted into current control input signals, which are then uploaded to the microcontroller. The current control signals can be encoded as the duty cycle of a square wave through PWM modulation and transferred to the H-bridge boards through the PWM pin of the microcontroller. The PWM current signals are then amplified by the H-bridge boards to control the current supply of each Helmholtz coil and thus each magnetic field component.

In experiments, the soft-magnet-based capsule robot is suspended in water in a cubic transparent container with a dimension of 100mm × 100mm × 100mm, which is placed inside the central workspace of the tri-axial Helmholtz coils. Two high-speed industrial cameras (OSG030-790UMTZ, YVSION Inc.) are used to record the front view and the top view of the dynamic robot rotation, which has a variable frame rate of 60-900 fps and a resolution of 640 × 480 pixels. In experiments to estimate torque by measuring time of rotation, the frame rate of 60, 310, 660, and 900 fps are used. Polar coordinate papers are put on the surface of the cubic container to confirm the robot orientation in the experiments of 3-DoF manipulation.

## Torque estimation from time of rotation

Since the magnetic torque generated on the soft magnet is very small (in the order of $10^{-5}$ N · m), the resolution of existing torque sensors (in the order of $10^{-4}$ N · m) is not sufficient for direct measurement of the torque. Therefore, an indirect method is used instead to estimate the magnetic torque from the rotation time of the soft magnet, which can be obtained from video recordings of high-speed cameras. Torque estimation from time is based on modeling of rotation dynamics of the soft-magnet robot. In experiments to validate theoretical models of magnetic torques, we need to measure the maximum magnetic torques about three principal axes of the elliptical soft magnet. It is known that the maximum magnetic torque about each axis is achieved when the external magnetic field is applied in the plane perpendicular to that axis and forms an angle of 45° with the long axis of the projected ellipse on the perpendicular plane. Therefore, the soft-magnet capsule robot is first orientated 45° from the applied field. Then the robot is released to rotate about a principal axis under a magnetic torque that varies with the orientation of the soft magnet. When the soft magnet rotates to approach the applied field, the magnetic torque decreases and comes to zero when it is aligned with the applied field. From Eq. (9), it is known that the magnetic torque during rotation is given by

$$\tau = \tau_{max} \sin(2\varphi) \quad (11)$$

where $\tau_{max}$ is the maximum torque, i.e. the torque at the initial orientation, and $\varphi$ is the angle between the long axis and the applied field. From Newton's law for rotation, the torque $\tau$ is related to the angular acceleration $\alpha$ by the moment of inertia $I$ about the rotation axis:

$$\tau = I\alpha \quad (12)$$

Let the elapsed rotation angle of the soft magnet be $\psi$, which is related with $\varphi$ by $\psi = 45° - \varphi$. Combining (11) and (12) gives the following equation:

$$\alpha = \frac{\tau_{max}}{I} \cos(2\psi) \quad (13)$$

Since the acceleration is the second order derivative of the rotated angle, Eq. (13) can be rewritten as

$$\ddot{\psi} = \frac{\tau_{max}}{I} \cos(2\psi) \quad (14)$$

which is a nonlinear second order differential equation. Introduce the intermediate variable $\dot{\psi}$, Eq. (14) can be simplified as

$$\dot{\psi} = \sqrt{\frac{\tau_{max}}{I} \sin(2\psi)} \quad (15)$$

Which can be solved by numerical integration given the initial and final condition of $\psi$.

$$\sqrt{\frac{\tau_{max}}{I}} t = \int_0^{\frac{\pi}{4}} \frac{1}{\sqrt{\sin(2\psi)}} d\psi = 1.311 \quad (16)$$

Then the relation between the maximum torque and the rotation time can be obtained as follows.

$$\tau_{max} = 1.719 \frac{I}{t^2} \quad (17)$$

where the inertial moment of the capsule robot (including the capsule shell and the soft magnet) is computed based on its geometry and density using the 3D CAD software (Solidworks, Dassault Systèmes). Once the rotation time is obtained from the video recordings by counting the number of corresponding frames, the maximum torque can be estimated from Eq. (17).

## Data availability
All the relevant data supporting the findings of this study are available within the paper and the Supplementary Information.

## Code availability
All the relevant code used to generate the results in this paper and the Supplementary Information is available upon request.

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

## Acknowledgements

The research work is financially supported by Grant 62203175 (H.W.) from the National Natural Science Foundation of China, Grant 2023A1515011553 (H.W.) from Guangdong Basic and Applied Basic Research Foundation, and Grant SL2022A04J01440 (H.W.) from Guangzhou Municipal Science and Technology Bureau.

## Author contributions

H.W. initiated and designed the project. H.W., J.C. and K.T. developed the theoretical models of anisotropic magnetization and magnetic torque of the soft magnet. J.C. analyzed the orientation stability and conducted experiments to validate theoretical models of magnetic torques. J.C. and K.T. built the tri-axial orthogonal Helmholtz coils and fabricated the soft magnet-based capsule robot. K.T. proposed the alternating open-loop orientation control strategy. Y.H. conducted experiments on 3-DoF orientation manipulation. H.W. analyzed the results, wrote the paper, and supervised the project.

## Competing interests

The authors declare no competing interests.
