## [Peer Review File · Nature Communications]

REVIEWER COMMENTS

Reviewer #1 (Remarks to the Author):

This paper primarily focuses on the full 3-DoF manipulation of an untethered small-scale robot. The proposed method involves replacing the regular permanent magnet with a soft magnet of anisotropic shape, enabling the achievement of full 3-DoF. This design is both interesting and attractive, presenting a promising application for future in-body control and manipulation schemes. However, there are some comments that need to be addressed.

1. The paper's writing requires improvement, and careful attention should be given to grammar and sentence structure.
2. The title "3-DoF orientation manipulation" is a bit deceiving. From a robotic perspective, the microrobot is always in a singular configuration (i.e. once it reaches an equilibrium, there is always an orientation that cannot be controlled). This is the reason why the authors have to do this multi-step open loop control. Whatever the orientation of the microrobot under a given magnetic field, there is always a direction along which a disturbance cannot be rejected (either unstable, in which case the microrobot flips to a new equilibrium, or "marginally stable" in which there is basically no counter-active torque to the disturbance). From a control perspective, this is a real problem and I don't think we can really talk about "3-DoF manipulation" in this case. Perhaps the authors can justify their view on this in more detail.
3. A sentence in the discussion says it all: "[...] 3-DoF orientation manipulation cannot be achieved using a single actuating magnetic field [...] all three DoFs cannot be simultaneously controlled." However this is exactly what n-DoF manipulation would mean, and the authors explicitly say here that they cannot achieve this.
4. Section 1 appears to be unrelated to the contributions of the paper and should be reduced or removed to maintain focus.
5. In Fig. 2i, the color bar in the middle figure displays incorrect units (rad instead of deg). This should be corrected to ensure accuracy.
6. The formulation and validation of eq. (10) do not appear to contribute significantly to the paper. It is derived from eq. (9) and doesn't add relevance or validity to the model.
7. Figure 4 seems overly complex and could be improved by presenting the results in a table format. Additionally, a simple schematic of the microrobot with its axes and external magnetic field for each case (a, b, and c) would be clearer.
8. In Section 2.4, the stability of the microrobot needs better characterization and description. Specify the amplitude of the perturbation angle and provide details on how its amplitude was

determined. The setup used for this characterization should also be described more precisely in terms of perturbation amplitude and direction.

9. Variable "b" is referenced on page 5, line 106, without prior introduction. Ensure all variables are defined before their usage.

10. Figure 3e should be presented as a separate figure since it is not related to figures 3a-d.

Reviewer #2 (Remarks to the Author):

The authors have developed an anisotropic soft magnet to enable 3-dof orientation manipulation with the application of external magnetic field. The authors have utilized three orthogonal Helmholtz coils to generate uniform magnetic fields in a desired direction to demonstrate their orientation manipulation. The authors have developed an open loop control method to manipulate the soft magnet from an initial orientation to the final orientation.

The authors also demonstrated their approach with both simulation and physical experiments.

While the capability of 3 dof orientation control of magnetic robot will enable various application, the control method the authors have lack both reliability and applicability.

Although the authors have identified the limitations of their approach in the discussion section, some of the limitations are significantly reduce its applicability.

1) The motivation of this approach as the authors have presented is to achieve 6 dof manipulation. The applicability of their approach cannot be justified if the authors approach cannot successfully enable 6 dof manipulation. Hence, the authors could enhance the results by providing results from 6 dof manipulation.

2) The other important limitation of this approach is the requirement of using soft magnet for 3 dof orientation control. The authors should provide some experimental data on the smallest robot where this approach can be applied successfully.

3) How feasible is this approach to make the control a closed loop rather than open loop?

4) "If the soft magnet is required to precisely stay at the target orientation, the two actuating magnetic fields

375 should be still alternatingly maintained along its longest and second longest axis to "lock" the soft magnet for

376 disturbance rejection." - Doesn't it hinder the applicability of this approach specially if we want to translate the robot?

5) "external observation is needed to determine the exact initial

orientation" - By "external observation" do the authors emphasize the need for perception system to determine the orientation of the robot?

Response to Review 1

The authors would like to thank the reviewer for their knowledgeable review and constructive comments. These comments have helped us improve the quality of the paper. We have addressed the issues brought up by the reviewer carefully and made corresponding modifications in the manuscript with the revised portions marked in red.

1. The paper's writing requires improvement, and careful attention should be given to grammar and sentence structure.

Response: We apologize for the errors and problems in grammar and sentence structures. We have carefully corrected the grammar errors and refined the language throughout the paper. We hope the readability of the revised manuscript is improved. The corrected part is marked in red in the revised manuscript.

2. The title "3-DoF orientation manipulation" is a bit deceiving. From a robotic perspective, the microrobot is always in a singular configuration (i.e. once it reaches an equilibrium, there is always an orientation that cannot be controlled). This is the reason why the authors have to do this multi-step open loop control. Whatever the orientation of the microrobot under a given magnetic field, there is always a direction along which a disturbance cannot be rejected (either unstable, in which case the microrobot flips to a new equilibrium, or "marginally stable" in which there is basically no counter-active torque to the disturbance). From a control perspective, this is a real problem and I don't think we can really talk about "3-DoF manipulation" in this case. Perhaps the authors can justify their view on this in more detail.

Response: We thank the reviewer for bringing up this issue. We would justify our view with the following points:

- (1) The proposed soft-magnet based actuation principle is fundamentally different from that of the permanent-magnet based actuation method. In permanent-magnet based method, no torque can be generated about its axis of magnetic moment (Section 1 in Results), resulting in only 2-DoF orientation manipulation. However, in our proposed soft-magnet based method, magnetic torque can be generated about all three directions (Section 2 and 3 in Results). The 3-DoF torque generation gives the capability of rotating the soft magnet (and the robot) in 3 DoFs. From the perspective of physical principle, we claim that the proposed soft-magnet based actuation method have the capability of 3-DoF orientation

manipulation.

- (2) It should be noted that the applied magnetic field itself is the control input to the soft-magnet robot rather than the environment where the robot works. In the proposed open-loop strategy, the orientation of the robot is controlled by varying the direction of the applied magnetic field. Therefore, when we discuss orientation manipulation of the robot, we cannot assume that the soft-magnet robot always stays in equilibrium under a fixed external magnetic field. On the contrary, we need to break the equilibrium of the soft magnet to generate the necessary torque for orientation manipulation.
- (3) The discussion on orientation stability of the soft magnet in a fixed applied field (Section 4 in Results) is only used to demonstrate the hierarchical tendency of the soft magnet to align its principal axes with the applied field. This is the foundation of the proposed open-loop control strategy that uses two alternating and orthogonal magnetic fields to guide the directions of the most stable (longest) and the second most stable (second longest) axes of the soft magnet. Again, it is noted that the magnetic torque is generated only when the applied field is misaligned with the principal axes (i.e. the soft magnet is NOT in the equilibrium state). In the paper, we state that the two orthogonal magnetic fields “lock” and guide the directions of the stable axes of the soft magnet during the open-loop control process because the settling time of the dynamic alignment of two stable axes with the applied field is very short (< 7 ms). However, we CANNOT say that the soft-magnet robot is controlled while staying in equilibrium.
- (4) Based on the proposed open-control strategy, the directions of the most stable (longest) and the second most stable (second longest) axes of the soft magnet are controlled by two orthogonal magnetic fields alternatingly. It is known that the 3-DoF orientation of a rigid body can be determined once directions of two orthogonal axes of the rigid body are determined. Therefore, we maintain that we achieved 3-DoF orientation manipulation of the soft-magnet robot.
- (5) On the other hand, if we are given the feedback of orientation of the soft-magnet robot in real time, a proper magnetic field can always be computed and applied to generate the desired torque for 3-DoF orientation control. In this closed-loop control scheme, we do not need multi-step control or alternating magnetic fields anymore. Furthermore, closed-loop control does not rely on the hierarchical tendency of the soft magnet to align its principal axes with the applied field. Therefore, the soft-magnet actuation system has the inherent capability of 3-DoF orientation manipulation that does not depend on certain control method. We favor the proposed open-loop control strategy because installing an orientation sensor system on the small untethered robot increases the complexity and cost of the system.
- (6) From a perspective of application, we can control the soft-magnet robot to reach any arbitrary 3-DoF orientation in $SO(3)$ (Section 5 in Results).
- (7) “3-DoF manipulation” or “6-DoF manipulation” are commonly used in the research community to refer to controlling the 3-DoF position (orientation) or 6-DoF pose of the small-scale robots. In this paper, we only investigated and demonstrated 3-DoF orientation manipulation although the soft-magnet actuation system has the capability of 6-DoF pose manipulation, which is now under investigation. Therefore, the title “3-DoF orientation manipulation” is proper in this context.

Based on the explanations above, we maintain that the title of “3-DoF orientation manipulation” is proper. We hope the clarification is satisfactory to the reviewer and we are open and happy to have further discussions with the reviewer if they have further questions or concerns.

3. A sentence in the discussion says it all: “[...] 3-DoF orientation manipulation cannot be achieved using a single actuating magnetic field [...] all three DoFs cannot be simultaneously controlled.” However this

is exactly what n-DoF manipulation would mean, and the authors explicitly say here that they cannot achieve this.

Response: We apologize for the misleading expression. This sentence actually means that 3-DoF orientation manipulation cannot be achieved **by controlling a single actuating magnetic field in an open-loop manner**. This sentence is used to express the necessity of designing the alternating and multi-step control strategy using **two orthogonal magnetic fields**. In this work, we investigate 3-DoF orientation manipulation from a general perspective of application. Any method can be said to be able to achieve 3-DoF orientation manipulation as long as it can control the small robot to reach any arbitrary 3-DoF orientation in $SO(3)$. Therefore, based on this general view, the proposed soft-magnet actuation method with alternating open-loop control strategy can achieve 3-DoF orientation manipulation. On the other hand, As stated in the response to question 2 above (point (5)), closed-loop control method can be used to achieve 3-DoF orientation manipulation with a single actuating magnetic field. However, closed-loop control increases the complexity and cost of the system and is out of the scope of this paper. In the revised manuscript, the proposed open-loop control method for 3-DoF orientation manipulation is clarified and the corresponding sentences are modified to eliminate misleading and confusing expressions.

4. Section 1 appears to be unrelated to the contributions of the paper and should be reduced or removed to maintain focus.

Response: Although we agree with the reviewer that this section is not the contribution of this paper and it should be reduced, we still consider this section necessary due to the following reasons.

- (1) In Section 1 in Results, a general introduction of magnetic actuation is provided to the audience not familiar with the topic by illustrating the basic principle and setup of magnetic actuation systems. It is hard for one to understand soft-magnet actuation without any prior knowledge of permanent-magnet actuation.
- (2) In addition, the fundamental formulas of magnetic force and torque is given in this section, which are the foundation of deriving models of magnetic torque in the soft-magnet actuation system.
- (3) Finally, this section explains why the permanent-magnet actuation system cannot achieve 3-DoF orientation manipulation and provides the motivation of developing the soft-magnet actuation system. Although the motivation of the paper is already given in the introduction part, a further explanation with a figure points out the deficiency of traditional magnetic actuation systems more clearly and highlights the importance of developing new magnetic actuation technologies.

Based on the reasons above, we decide to keep but reduce this section. In the revised manuscript, Section 1 in Results is reduced to maintain the focus.

5. In Fig. 2i, the color bar in the middle figure displays incorrect units (rad instead of deg). This should be corrected to ensure accuracy.?

Response: We apologize for the error in this figure. In the revised manuscript, the unit in the color bar has been corrected into “degree”.

6. The formulation and validation of eq. (10) do not appear to contribute significantly to the paper. It is derived from eq. (9) and doesn't add relevance or validity to the model.

Response: Equation (10) gives the relation of maximum magnetic torques about three principal axes and thus shows the capability of torque generation about three axes. It is found that the largest torque can be generated about **b**-axis (second longest axis) and it is exactly the sum of maximum torques about the remaining two

axes. **This is one of the most important conclusions and contributions of this paper.** It is noted that the purpose of this paper is not only to just demonstrate the application of the soft-magnet actuation for 3-DoF orientation manipulation of small robots, but also to study and illustrate the basic physical principle of this new magnetic actuation method since it has never been systematically investigated in existing research. Equation (10) is an interesting conclusion that shows the unequal capability of torque generation about different axes in the soft-magnet actuation system, which is quite different from the case of traditional permanent-magnet system that has equal torque generation capability about two axes (permanent-magnet system cannot generate torque about the third axis of magnetic moment). Equation (10) is also a useful conclusion that can help optimize open-loop orientation control. For example, the duty ratio of the alternating orthogonal magnetic fields can be tuned according to the torque generation capability about different axes in order to make the orientation manipulation more fluent. Although the property described by Equation (10) is not explicitly applied in 3-DoF orientation manipulation in this paper, we consider it as an important contribution of this work itself and will continue to study its impact and application in the future work.

Furthermore, experimental validation of Equation (10) implicitly shows the correctness of analytical modeling of the anisotropic magnetization and magnetic torques of the soft magnet.

Based on the explanation above, we maintain to reserve the formulation and discussion of Equation (10). In the revised manuscript, we further clarified the importance of Equation (10) and modified the related part to add relevance to the focus of the paper.

7. Figure 4 seems overly complex and could be improved by presenting the results in a table format. Additionally, a simple schematic of the microrobot with its axes and external magnetic field for each case (a, b, and c) would be clearer.

Response: We apologize for the unclear illustration in Figure 4. In the revised manuscript, we have modified Figure 4 (Figure 5 in the revised manuscript). Specifically, we modified the figure to match schematics and photos of the soft magnet in each case such that readers can easily correlate and understand them. We also explicitly listed each state of the robot above photos and schematics, i.e. initial state, perturbed state, and final state. Furthermore, we drew dashed elliptical outlines of the soft magnet inside the capsule robot in order to visualize the invisible soft magnet in the photos. We also added a brief description in the text to help readers understand the figure.

8. In Section 2.4, the stability of the microrobot needs better characterization and description. Specify the amplitude of the perturbation angle and provide details on how its amplitude was determined. The setup used for this characterization should also be described more precisely in terms of perturbation amplitude and direction.

Response: As long as the perturbation angle is less than 90° , the rotational behaviors of the soft magnet in stable, marginally stable, or unstable state are the same as those shown in Figure 5. If the perturbation angle is larger than 90° , the soft magnet would rotate to the next equilibrium. In the context of orientation stability analysis, we assume the perturbation is small ($< 90^\circ$). Therefore, the perturbation amplitude can be any value below 90° . In addition, the orientation stability of the soft magnet does not depend on the perturbation direction due to the symmetry of the anisotropic soft magnet (e.g. tri-axial ellipsoid). With this said, to maintain consistency, the perturbation amplitude is set as 45° and the perturbation direction is set as positive direction based on right-hand rule in the experiments shown in Fig. 5. In the revised manuscript, we added descriptions of perturbation amplitudes and direction in the text and figure legend.

9. Variable "b" is referenced on page 5, line 106, without prior introduction. Ensure all variables are defined before their usage.

Response: In the revised manuscript, we have added definition of variable “B” and “m” where they are referenced.

10. Figure 3e should be presented as a separate figure since it is not related to figures 3a-d.

Response: In the revised manuscript, Figure 3 e is presented as a separate figure (Figure 4) in the revised version.

Response to Review 2

We thank the reviewer for their knowledgeable review and thoughtful comments, many of which have been very helpful for us to improve the paper. We have modified the manuscript carefully and the revised portions are marked in red in the manuscript. Below are our responses to the reviewer’s questions and comments.

1. The motivation of this approach as the authors have presented is to achieve 6 dof manipulation. The applicability of their approach cannot be justified if the authors approach cannot successfully enable 6 dof manipulation. Hence, the authors could enhance the results by providing results from 6 dof manipulation.

Response: We thank the reviewer for initiating the discussion on 6-DoF manipulation. We would like to clarify the following points.

- (1) First of all, we would clarify that the scope of this paper is limited to 3-DoF orientation manipulation using an anisotropic soft magnet while it lays the foundation for full 6-DoF magnetic manipulation. It is noted that this paper is not fully application-oriented. Since the proposed soft-magnet based actuation method has never been systematically investigated before and its working principle is not well understood and is very different from and much more complicated than the traditional permanent-magnet based actuation method, another purpose of the paper is to explain the fundamental physical principle of the new magnetic actuation method besides demonstrating its application in 3-DoF orientation manipulation.
- (2) The paper is self-contained with theoretical modeling of magnetic torque and experimental demonstration of feasibility of precise 3-DoF orientation manipulation of small robots. Since the magnetic force on an anisotropic soft magnet and its translational dynamics are also much more complicated than the permanent-magnet system, we do not have enough space in this paper to include theoretical modeling and experimental validation of 3-DoF translational manipulation using the soft magnet. 3-DoF translational manipulation and 6-DoF manipulation is under investigation and we will publish it in another paper.
- (3) Although 6-DoF manipulation is not experimentally demonstrated yet in this work, the findings in this paper solved the greatest challenge for 6-DoF manipulation because the bottleneck of existing magnetic manipulation technologies is their inability of 3-DoF orientation manipulation rather than translation manipulation. From a theoretical perspective, magnetic forces can always be generated in all three directions as long as proper magnetic gradients are applied to the magnetic object (whether it is a soft

magnet or a permanent magnet). Therefore, the proposed soft-magnet based actuation method holds the capability of full 6-DoF manipulation.

- (4) To demonstrate the potential of 6-DoF manipulation of the proposed soft-magnet based actuation method, we would provide some preliminary theoretical work on modeling of magnetic forces and a promising control method for 6-DoF manipulation that builds upon the proposed orientation control method. This preliminary work is given here and added to the supplementary materials of the paper (Supplementary note 2 and Supplementary Fig. 4 - 5).

Magnetic force generation in 3 DoFs

The magnetic force on a magnetic dipole depends on the spatial gradient of the local magnetic flux density, which is given below

$$= \nabla(\cdot) = \begin{bmatrix} \frac{\partial}{\partial x} & \frac{\partial}{\partial y} & \frac{\partial}{\partial z} \end{bmatrix}^T \begin{bmatrix} \frac{\partial}{\partial x} & \frac{\partial}{\partial y} & \frac{\partial}{\partial z} \\ \frac{\partial}{\partial x} & \frac{\partial}{\partial y} & \frac{\partial}{\partial z} \\ \frac{\partial}{\partial x} & \frac{\partial}{\partial y} & \frac{\partial}{\partial z} \end{bmatrix} \quad (R1)$$

Now replace the magnetic dipole with an anisotropic soft magnet, with **a**-, **b**-, and **c**-axis of the soft magnet aligned with the **x**-, **y**-, and **z**-axis of the world coordinate system (WCS) as shown in Fig. A (a). The soft magnet's magnetization is determined by Equation (3) - (7) in the manuscript. Combining these equations with Equation (R1), we can obtain magnetic force on the soft magnet in an external magnetic field :

$$= \frac{1}{\mu_0} (\cdot \nabla) = \frac{1}{\mu_0} \begin{bmatrix} \frac{\partial}{\partial x} & \frac{\partial}{\partial y} & \frac{\partial}{\partial z} \\ \frac{\partial}{\partial x} & \frac{\partial}{\partial y} & \frac{\partial}{\partial z} \\ \frac{\partial}{\partial x} & \frac{\partial}{\partial y} & \frac{\partial}{\partial z} \end{bmatrix} \quad (R2)$$

where V is the volume of the soft magnet and μ_0 is the permeability in vacuum. From Equation (R2), to apply a desired force to the soft magnet, we need to provide both an appropriate magnetic field and an appropriate magnetic field spatial gradient. Equation (R2) also shows that it is feasible to generate magnetic forces on the soft magnet in three directions for 3-DoF translation control.

Fig. A. Magnetic force on soft magnet in a magnetic field with spatial gradient. **a.** Schematic of anisotropic magnetization and magnetic torque of a tri-ellipsoidal soft magnet. The anisotropic magnetization of the soft magnet leads to separation of the direction of magnetization vector from the applied field, resulting in a non-zero magnetic torque. **b.** Alignment of an ellipsoidal soft magnet's **a**-

axis with the direction of magnetic field due to the torque. **c.** Schematic of 3-DoF magnetic force on the soft magnet subjected to both magnetic field and magnetic field gradient ∇ .

The orthogonal Helmholtz coils used in the paper can only generate a uniform magnetic field but lacks the capability to produce magnetic field spatial gradients. Therefore, we need to design a new magnetic field generation setup, i.e., the Helmholtz-Maxwell combination coils. The Helmholtz-Maxwell coils consist of three pairs of Helmholtz coils and three pairs of Maxwell coils, both of which are installed orthogonally. A uniform magnetic field can be generated in any direction by Helmholtz coils that runs the currents in the same direction in pairs of coils, while three magnetic field gradients, namely $\frac{\partial B}{\partial x}$, $\frac{\partial B}{\partial y}$, and $\frac{\partial B}{\partial z}$, can be provided by Maxwell coils that runs the currents in the opposite direction in pairs of coils. In the working space of the setup, the magnitude of $\frac{\partial B}{\partial x}$, $\frac{\partial B}{\partial y}$, and $\frac{\partial B}{\partial z}$ are negligible compared to the magnitude of B_x , B_y , and B_z . Therefore, Equation (1) can be simplified to

$$\mathbf{F} = \mu_0 \mathbf{m} \nabla B \quad (R3)$$

As shown in Fig. A (a) and (b), when B is applied in an arbitrary direction without a magnetic field spatial gradient, the soft magnet experiences a torque that causes its \mathbf{a} -axis to align with the direction of the magnetic field according to the analysis of orientation stability in the paper. Once aligned, the magnetic moment of the soft magnet under B is given by:

$$\mathbf{m} = \frac{\mu_0 \mathbf{a}}{B} \quad (R4)$$

Substitute (R4) into (R3), the following equation of magnetic force is obtained:

$$\mathbf{F} = \mu_0 \frac{\mathbf{a}}{B} \left[\frac{\partial B}{\partial x} \mathbf{i} + \frac{\partial B}{\partial y} \mathbf{j} + \frac{\partial B}{\partial z} \mathbf{k} \right]^T \quad (R5)$$

By analyzing Equation (R5), we can draw the following conclusion: Knowing that \mathbf{a} -axis of the soft magnet aligns with the direction of B , we can control the magnetic force on the soft magnet by regulating the terms $\frac{\partial B}{\partial x}$, $\frac{\partial B}{\partial y}$ and $\frac{\partial B}{\partial z}$ through the Helmholtz-Maxwell combination coils, as shown in Figure.

A (c).

Control method for 6-DoF manipulation

In the following section, we will introduce a control strategy that we are currently conceptualizing for 6-DoF manipulation of the soft-magnet robot. This 6-DoF control strategy is built upon the open-loop 3-DoF orientation control method proposed in this paper.

Fig. B (a), (b), and (c) illustrate translation control in three directions. Taking Fig. 2. (a) as an example, a magnetic field B_x is applied in \mathbf{x} -axis to align the \mathbf{a} -axis of the soft magnet and magnetic moment \mathbf{m} with the field by activating the Helmholtz coils along the \mathbf{x} -axis. Then activate the Maxwell coils along the \mathbf{x} -axis to generate

while maintaining B_x . According to Equation (R5), the force can be expressed as $\mathbf{F} = \mu_0 \frac{\mathbf{a}}{B} \left[\frac{\partial B}{\partial x} \mathbf{i} + 0 \mathbf{j} + 0 \mathbf{k} \right]^T$. Therefore, F_x does not depend on B_y and B_z and we can control the magnetic

force in \mathbf{x} -axis by tuning only $\frac{\partial B}{\partial x}$. Similarly, we can control the magnetic force in \mathbf{y} -axis and \mathbf{z} -axis by

tuning and , respectively, as depicted in Fig. B (b) and (c). Finally, a position sensor is needed to provide 3-DoF position of the soft-magnet robot for feedback (closed-loop) translation control.

Fig. B. Translation control and 6-DoF manipulation of an anisotropic soft magnet. a. Translation control in x-axis. **b.** Translation control in y-axis. **c.** Translation control in z-axis. **d.** 6-DoF manipulation of the soft magnet with decomposed translation and orientation control.

6-DoF manipulation can be decomposed into 3-DoF translation and 3-DoF rotation. First, the soft-magnet robot is translated in three directions one at a time. When shifting among the three translation directions, we need to rotate the soft magnet to align its a-axis with the translation direction. For instance, as shown in Fig. B (d), when the soft-magnet robot is switched from x-axis translation to y-axis translation, it needs to rotate 90 ° about z-axis using the proposed open-loop orientation control method to align the longest a-axis with the heading direction. Once the soft-magnet robot reaches the target position through translation, we can utilize the orientation control method proposed in the paper to manipulate its orientation. The whole process of 6-DoF manipulation is illustrated in Fig. B (d).

2. The other important limitation of this approach is the requirement of using soft magnet for 3 dof orientation control. The authors should provide some experimental data on the smallest robot where this approach can be applied successfully.

Response: We have conducted additional experiments using smaller soft magnets and smaller capsule robots. It is shown that the proposed actuation method works well for soft magnets with a size down to that of commercial capsule endoscopes ($\varnothing 14 \text{ mm} \times 28 \text{ mm}$) and capsule robots with a size down to $10 \times 1.2 \times 4.8 \text{ mm}$. In the revised manuscript, experimental data of 3-DoF orientation manipulation of smaller soft-magnet robots are added to the supplementary materials (Supplementary Figure 3).

3. How feasible is this approach to make the control a closed loop rather than open loop?

Response: The closed-loop control method is feasible if an orientation sensor is installed on the robot to provide 3-DoF orientation information in real time. The block diagram of open-loop control method is given below (Fig. C).

Fig. C. Closed-loop control scheme for orientation manipulation of soft-magnet robot.

In the closed-loop control, a desired torque is determined by the proportional-integral-derivative (PID) controller based on the difference between the desired orientation and measured orientation of the soft-magnet robot. Then the desired direction of magnetic field is computed from the inverse model of magnetic torque (i.e. the inverse of Equation (9) in the manuscript). Finally, a desired current supply to the Helmholtz coils can be determined from a pre-calibrated inverse model, which enables the Helmholtz coils to generate the desired magnetic torque for 3-DoF orientation manipulation.

Although the closed-loop control is feasible, there are pros and cons for it. In closed-loop control, we can use a single magnetic field for robot control rather than the complicated alternating and multi-step control strategy using two orthogonal magnetic fields as in the open-loop control. However, closed-loop control requires an additional orientation sensor and a wireless communication module installed on the small robot, which increases the complexity and cost of the system. Furthermore, the control loop shown in Fig. C should be updated at a very high frequency to ensure stability and accuracy of orientation manipulation since the rotation dynamics of the soft-magnet robot is very fast. Therefore, we favor the proposed open-loop control method for its simplicity and minimum requirement for hardware and computational resources.

4. "If the soft magnet is required to precisely stay at the target orientation, the two actuating magnetic fields should be still alternatingly maintained along its longest and second longest axis to "lock" the soft magnet for disturbance rejection." - Doesn't it hinder the applicability of this approach specially if we want to translate the robot?

Response: The proposed alternating open-loop control method for orientation manipulation does not hinder translation of the robot. We have proposed a preliminary 6-DoF manipulation strategy that builds upon the current 3-DoF orientation control method, which is provided in the response to question 1 above and also added to supplementary materials. It is seen that the magnetic force for translation control depends on both

the magnetic field and magnetic field gradient (Equation R2). While magnetic field needs to be controlled for orientation manipulation, we can still control the magnetic field gradient for translation control without affecting the orientation control since the magnetic torque depends only on the magnetic field itself. Therefore, the orientation control and translation control can be decomposed and the proposed orientation control method is compatible to translation control and full 6-DoF manipulation of the soft-magnet robot. We thank the reviewer for initiating the discussion on applicability of our proposed orientation control method.

5. "external observation is needed to determine the exact initial orientation" - By "external observation" do the authors emphasize the need for perception system to determine the orientation of the robot?

Response: The magnetic torque can possibly align either the positive or the negative longer axis of the soft magnet with the applied field, depending on whichever end is closer to the applied field. In other words, if the angle between the positive longer axis and the applied field is less than 90° , then the soft magnet aligns its positive axis with the applied field. Otherwise, the negative axis aligns with the applied field. Therefore, this problem can possibly cause the reverse alignment of the soft magnet with the desired initial orientation when a magnetic field is applied to the soft magnet initially. The true resulting orientation is one of the four possible orientations (See Supplementary Figure 2). Since the initial true orientation of the soft magnet needs to be known for later open-loop orientation control, some external observation is needed to determine the true initial orientation from the four possibilities. "External observation" can be either a camera, a low-accuracy orientation sensor, or even human observation. It is noted that this "observation" does not need to be accurate measurement of orientation but any perception method that can distinguishes the four possible orientations is feasible. It is also noted that external observation or perception of orientation is only needed at the initial time and later 3-DoF orientation manipulation is performed in a fully open-loop fashion that does not need any orientation sensing. In the revised manuscript, we have further clarified what kind of "external observation" is needed.

REVIEWER COMMENTS

Reviewer #1 (Remarks to the Author):

This article, compared to traditional methods that use permanent magnets, employs soft magnetic materials to control small-scale robots. This innovative approach demonstrates the feasibility of 3DoF manipulation and lays the foundation for 6DoF manipulation. While the ideas presented in this article still have limitations and are influenced by some related parameters, they have made significant advances and hold great promise for future research and applications. In the revised version, the authors have further clarified or detailed many of the unclear or ambiguously expressed parts and made corrections to wrong expressions, improving the readability and logic of the article. Additionally, there are still some minor comments that I wish to discuss with the authors:

The article mentions cylinders or cuboids. Could a brief discussion of this situation also be included?

It is suggested to place Figure 4 in the supplementary information.

In Figure 5, the schematic diagrams on the right, due to their drawing style, have been misunderstood as different soft magnets in the same field, which could easily mislead readers. It's better to divide into different parts by lines.

In Figure 7c, there is a brief explanation of the impact of different frequencies on the results. Could the authors provide a more detailed explanation in the article? Are there upper and lower limits for frequency?

In the discussion section of the article, there is mention of the potential application in the GI tract. Is there an evaluation and discussion of the manipulation performance in the environmental flow field? Because it will bring in a lot of disturbance.

Reviewer #2 (Remarks to the Author):

The authors have addressed all the comments the reviewer had.

Response to Review 1

The authors would like to thank the reviewer for their insightful review and constructive comments. These comments have helped us improve the quality of the paper. We have addressed the remaining concerns brought up by the reviewer carefully and made corresponding modifications in the manuscript with the revised portions marked in red.

1. The article mentions cylinders or cuboids. Could a brief discussion of this situation also be included?

Response: We thank the reviewer for bringing up this issue. As is shown in Equation (7), the magnetization \mathbf{M} (or magnetic moment \mathbf{m}) of the soft magnet only depends on the demagnetization factors n_a , n_b , and n_c in three principal axes. Although these demagnetization factors depend on the geometric dimensions of the soft magnet, the modeling of magnetic torques and rotational dynamics provided in the paper is valid for all shapes of soft magnets. Whatever shape of the soft magnet is used, the behavior of orientation manipulation would be the same as long as their demagnetization factors are the same. The computation of demagnetization factors for tri-axial ellipsoids, elliptical cylinders, and cuboids are provided in Supplementary Note 1. In the paper, we take tri-axial ellipsoid as an example to present the theoretical modeling schematically while we actually use elliptical cylinders for experiments. In practice, we recommend using elliptical cylinders or cuboids (prisms) for their simplicity of machining. In the revised manuscript, we have added a discussion on the use of elliptical cylinders and cuboids for soft-magnet manipulation in Section 2 in Results.

2. It is suggested to place Figure 4 in the supplementary information.

Response: In the revised manuscript, we have moved Figure 4 to the supplementary information. Thanks for the suggestion.

3. In Figure 5, the schematic diagrams on the right, due to their drawing style, have been misunderstood as different soft magnets in the same field, which could easily mislead readers. It's better to divide into different parts by lines.

Response: We apologize for the misleading presentation of Figure 5 (Fig. 4 in the revised manuscript). In the revised manuscript, we have divided the schematic diagrams of soft magnets in different states by dashed lines.

4. In Figure 7c, there is a brief explanation of the impact of different frequencies on the results. Could the authors provide a more detailed explanation in the article? Are there upper and lower limits for frequency?

Response: The frequency of the alternating magnetic control is 10 Hz in the experiments shown in Fig. 7 (Fig. 6 in the revised manuscript), which is sufficient to rotate the soft-magnet robot smoothly. The frequency of magnetic control cannot be too low for two reasons:

(1) Since the proposed open-loop control strategy rotates the soft-magnet robot in two alternating stages, i.e., controlling the direction of the longest axis and controlling the rotation about the longest axis, low frequency of the alternating magnetic control results in non-smooth orientation manipulation, which is not desired for continuous robot control. Supplementary Movie 9 (Section 2) show the orientation manipulation at 2 Hz, where the soft-magnet robot is rotated in a non-smooth manner.

(2) It is shown in Fig. 4 (orientation stability) that the longest axis of the soft magnet always tends to align with the applied magnetic field. Lower frequency of magnetic control allows for a longer period of rotation about the longest axis, during which potential disturbances might deflect the longest axis so an undesired magnetic torque is generated to drive the longest axis towards the direction of \mathbf{B}_i^r , which is not desired (the longest axis should be pointed to the direction of \mathbf{B}_i^a) and might cause instability of orientation manipulation. Although this issue also exists in the case with high frequency of magnetic control, the potential wrong deflection of the longest axis towards \mathbf{B}_i^r lasts for a shorter time and it can be quickly corrected by \mathbf{B}_i^a in the high-frequency case. Supplementary Movie 9 (Section 1) shows the orientation manipulation at 0.5 Hz, where the soft-magnet robot continues to rotate and fails to stabilize at the target orientation.

Theoretically, there is no upper limit of the frequency of magnetic control. However, the inductance effect of the Helmholtz coils cannot be neglected at high frequencies. At high frequencies, although the voltage input to the coils is a square wave, the current and the resulting magnetic field become approximately constant with a magnitude that is determined by the duty cycle. This is exactly the same as the principle of the pulse width modulation (PWM). Therefore, the alternating application of \mathbf{B}_i^r and \mathbf{B}_i^a become a constant application of $(1 - a)\mathbf{B}_i^r$ and $a\mathbf{B}_i^a$, where a is the duty cycle of \mathbf{B}_i^a . The final effect of the magnetic control at high frequencies is equivalent to applying a constant magnetic field $(1 - a)\mathbf{B}_i^r + a\mathbf{B}_i^a$, which degenerates to controlling a soft magnet with a single magnetic field. Since it is shown in the paper that a single magnetic field cannot fully control the 3-DoF orientation of the soft magnet, there indeed exists an upper limit of frequency beyond which the proposed control strategy fails. Supplementary Movie 10 shows the orientation manipulation at 10 kHz, where the soft magnet aligns its longest axis with the direction of the superimposed magnetic field and fails to reach the target orientation.

In summary, we have performed experiments to explore the lower and upper limits of feasible frequencies of the alternating magnetic control method. It is shown that the lower limit for smooth orientation manipulation is 2 Hz and the lower limit for stable manipulation is 0.5 Hz. The upper limit for successful 3-DoF orientation is around 10 kHz. In the revised manuscript, we have added a detailed explanation of the upper and lower limits of the frequency of magnetic control in Section 5 in Results.

5. In the discussion section of the article, there is mention of the potential application in the GI tract. Is there an evaluation and discussion of the manipulation performance in the environmental flow field? Because it will bring in a lot of disturbance.

Response: We thank the reviewer for pointing out this issue. Although this paper is not mainly devoted to the realistic application of the proposed soft-magnet actuation strategy, we agree that an evaluation of the impact of flow disturbance on soft-magnet robot manipulation can help demonstrate the applicability of the proposed method. We conducted an additional experiment to validate our soft-magnet actuation method using a water circulation system (Supplementary Figure 5), where the water in the tank to suspend the capsule robot is extracted by a water pump and meanwhile pumped back to the tank. The water circulation causes a flow disturbance to the orientation manipulation of the capsule robot. The flow rate is set to be 100 mL/min (The volume of liquid in the tank is about 750mL), which is higher than that of the fluid flow in our GI tract. Results show that the flow disturbance has negligible influence on 3-DoF orientation manipulation of the soft-magnet robot (Supplementary Figure 6 and Supplementary Movie 8). In the revised manuscript, we have added a discussion of the manipulation performance in the environmental flow field.

Response to Review 2

We thank the reviewer for their knowledgeable and very efficient review. The comments have been very helpful for us to improve the paper.

REVIEWERS' COMMENTS

Reviewer #1 (Remarks to the Author):

We believe that the authors have effectively addressed the questions and clarified the previously confusing points in the previous revised version, offering readers a clearer understanding of this paper. Additionally, the authors have corrected minor errors and adjusted potentially misleading expressions, which has significantly enhanced the overall readability of the manuscript. The interesting and innovative concept is now well-articulated in this revised version of the manuscript.

Response to Review 1

The authors would like to thank the reviewer for their insightful and very efficient review and constructive comments. These comments have helped us improve the quality of the paper. We appreciate their persistent interest in our work and we are happy to hear that our earlier response addressed the concerns of the reviewer.